# Geodesic Optimization for Predictive Shift Adaptation on EEG data

**Apolline Mellot**,[*] **Antoine Collas**[*]
Inria, CEA, Université Paris-Saclay
Palaiseau, France
`apolline.mellot@inria.fr`
`antoine.collas@inria.fr`

**Sylvain Chevallier**
TAU Inria, LISN-CNRS,
University Paris-Saclay, France.
`sylvain.chevallier@`
`universite-paris-saclay.fr`

**Alexandre Gramfort**
Inria, CEA, Université Paris-Saclay
Palaiseau, France
`alexandre.gramfort@inria.fr`

**Denis A. Engemann**
Roche Pharma Research and Early Development,
Neuroscience and Rare Diseases,
Roche Innovation Center Basel,
F. Hoffmann–La Roche Ltd., Basel, Switzerland.
`denis.engemann@roche.com`

## Abstract

Electroencephalography (EEG) data is often collected from diverse contexts involving different populations and EEG devices. This variability can induce distribution shifts in the data $X$ and in the biomedical variables of interest $y$, thus limiting the application of supervised machine learning (ML) algorithms. While domain adaptation (DA) methods have been developed to mitigate the impact of these shifts, such methods struggle when distribution shifts occur simultaneously in $X$ and $y$. As state-of-the-art ML models for EEG represent the data by spatial covariance matrices, which lie on the Riemannian manifold of Symmetric Positive Definite (SPD) matrices, it is appealing to study DA techniques operating on the SPD manifold. This paper proposes a novel method termed Geodesic Optimization for Predictive Shift Adaptation (`GOPSA`) to address test-time multi-source DA for situations in which source domains have distinct $y$ distributions. `GOPSA` exploits the geodesic structure of the Riemannian manifold to jointly learn a domain-specific re-centering operator representing site-specific intercepts and the regression model. We performed empirical benchmarks on the cross-site generalization of age-prediction models with resting-state EEG data from a large multi-national dataset (HarMN-qEEG), which included 14 recording sites and more than 1500 human participants. Compared to state-of-the-art methods, our results showed that `GOPSA` achieved significantly higher performance on three regression metrics ($R^2$, MAE, and Spearman's $\rho$) for several source-target site combinations, highlighting its effectiveness in tackling multi-source DA with predictive shifts in EEG data analysis. Our method has the potential to combine the advantages of mixed-effects modeling with machine learning for biomedical applications of EEG, such as multicenter clinical trials.

## 1  Introduction

Machine learning (ML) has enabled advances in the analysis of complex biological signals, such as magneto- and electroencephalography (M/EEG), in diverse applications including biomarker

---

[*]Equal contribution.

38th Conference on Neural Information Processing Systems (NeurIPS 2024).

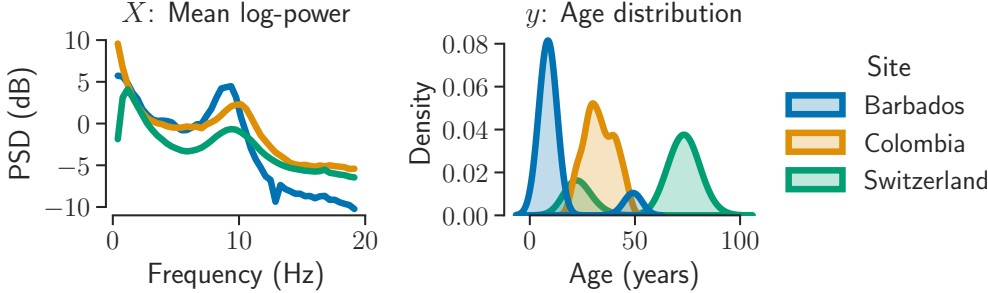

Figure 1: **Joint shift in $X$ and $y$ distributions on the HarMNqEEG dataset [33].** Subset of mean PSDs (**A**) and age distributions (**B**) from three recording sites used for the empirical benchmarks.

exploration [58, 20, 60] or developing Brain-Computer Interface (BCI) [57, 15, 1, 2]. However, a major challenge in applying ML to these signals arises from their inherent variability, a problem commonly referred to as dataset shift [12]. In the case of M/EEG data this variability can be caused by differences in recording devices (electrode positions and amplifier configurations), recording protocols, population demographics, and inter-subject variability [36, 14, 23, 29]. Notably, shifts can occur not only in the data $X$ but also in the biomedical variable $y$ we aim to predict, further complicating the use of ML algorithms.

Riemannian geometry has significantly advanced EEG data analysis by enabling the use of spatial covariance matrices as EEG descriptors [5, 4, 6, 38, 31, 35, 33, 48, 17, 56]. In [4, 6], the authors introduced a classification framework for BCI based on the Riemannian geometry of covariance matrices. These methods classify EEG signals directly on the tangent space using the Riemannian manifold of symmetric positive definite (SPD) matrices ($\mathbb{S}_d^{++}$), effectively capturing spatial information. More recently, [47, 48, 7] extended this framework to regression problems from M/EEG data in the context of biomarker exploration. Furthermore, [47, 48] proved that Riemannian metrics lead to regression models with statistical guarantees in line with log-linear brain dynamics [10] and are, therefore, well-suited for neuroscience applications. Across various biomarker-exploration tasks and datasets, recent work has shown that Riemannian M/EEG representations offer parameter-sparse alternatives to non-Riemannian deep learning architectures [13, 40, 17].

Domain adaptation addresses the challenges posed by differences in data distributions between source and target domains, e.g., when data are recorded with different cameras in computer vision [55], different writing styles in natural language processing [32], or varying sensor setups in time series analysis [22]. In particular, DA considers *target shift* where the shift is in the outcome variable $y$. For classification it means source and target data share the same labels but in different proportions [34]. Target shift is also frequent in the context of multicenter neuroscience studies, as the studied population of one site may vary significantly from the studied population of another site (cf. Figure 1). To tackle various sources of variability in neurophysiological data like EEG, there is a need for a DA approach that can deal with a joint shift in $X$ and $y$.

**Related work** In [62], the authors addressed DA for EEG-based BCI using re-centering affine transformation of covariance matrices (Section 2) to align data from different sessions or human participants, improving classification accuracies. Yair et al. [59] extended this with parallel transport showing its effectiveness in EEG analysis, whereas, Peng et al [43] introduced a domain-specific regularizer based on the Riemannian mean. Notably, this parallel transport approach reduces to [62] when the common reference point is the identity. In a deep learning context, Kobler et al. [31] proposed to do a per-domain online re-centering which can be seen as a domain specific Riemannian batch norm. Going beyond re-centering, Riemannian Procrustes Analysis (RPA) [45] was proposed for EEG transfer learning, using three steps: mean alignment, dispersion matching, and rotation correction. However, the rotation step is unsuitable for regression problems and RPA adapts only a single source to a target domain. Recently, [36] demonstrated the benefits of re-centering for regression problems, showing improvements in handling task variations in MEG and enhancing across-dataset inference in EEG.

On the other hand, mixed-effects models (or multilevel models) have been successfully used to tackle data shifts in $X$ and $y$ [16, 25]. In biomedical data, mixed-effects models are crucial due to the presence of common effects, such as disease status and age. Riemannian mixed-effects models have been used to analyze observations on Riemannian manifolds, accommodating individual trajectories with mixed effects at both group and individual levels [30, 49, 50]. These models adapt a base point on the manifold for each data point and utilize parallel transport for this adaptation, which is necessary for accurate trajectory modeling. However, they differ significantly from the problem we address in this work. Notably, the input data $X$ are covariates (e.g., age or disease status) which belong to a Euclidean space and the variables $y$ to predict belong to the manifold (e.g., MRI diffusion tensors on $\mathbb{S}_d^{++}$) which is the opposite of the paper's studied problem. This distinction is critical as it highlights that while both methods use the geometry of Riemannian manifolds, the nature of the predicted variables and the type of data used differ from existing Riemannian mixed-effects models.

**Contributions** In this work, we address the challenging problem of multi-source domain adaptation with predictive shifts on the SPD manifold, focusing on distribution shifts in both the input data $X$ and the variable to predict $y$. We propose a novel method called Geodesic Optimization for Predictive Shift Adaptation (GOPSA). It enables mixed-effects modeling by jointly learning parallel transport along a geodesic for each domain and a global regression model common to all domains, with the assumption that the mean $\bar{y}_{\mathcal{T}}$ of the target domain is known. GOPSA aims to advance the state of the art by: *(i)* addressing shifts in both covariance matrices and the outcome variable $y$, *(ii)* being tailored for regression problems, and *(iii)* being a multi-source test-time domain adaptation method, meaning that once trained on source data, it can generalize to any target domain without requiring access to source data or retraining a new model.

We first introduce in Section 2 how to do regression from covariance matrices on the Riemannian manifold of $\mathbb{S}_d^{++}$. We also interpret classical learning methods on $\mathbb{S}_d^{++}$ from heterogeneous domains as parallel transports combined with Riemannian logarithmic mappings. This leads us to GOPSA in Section 3, which learns to parallel transport each domain, with algorithms at train and test times. Finally, in Section 4, we apply GOPSA as well as different baselines on simulated data and the HarMNqEEG dataset.

**Notations** Vectors and matrices are represented by small and large cap boldface letters respectively (e.g., $\boldsymbol{x}$, $\boldsymbol{X}$). The set $\{1, ..., K\}$ is denoted by $[\![1, K]\!]$. $\mathbb{S}_d^{++}$ and $\mathbb{S}_d$ represent the sets of $d \times d$ symmetric positive definite and symmetric matrices. uvec : $\mathbb{S}_d \to \mathbb{R}^{d(d+1)/2}$ vectorizes the upper triangular part of a symmetric matrix. Frobenius and 2-norms are denoted by $\|.\|_F$ and $\|.\|_2$, respectively. $\triangleq$ defines the left part of the equation as the right part. The transpose and Euclidean inner product operations are represented by $\cdot^\top$. $\mathbf{1}_n \triangleq [1, \ldots, 1]^\top$ denotes an $n$-dimensional vector of ones. For a loss function $\mathcal{L}$, $\nabla \mathcal{L}$ and $\mathcal{L}'$ denote its gradient and derivative. We consider $K$ labeled source domains, each consisting of $N_k$ covariance matrices, along with their corresponding outcome values, denoted by $\{(\boldsymbol{\Sigma}_{k,i}, y_{k,i})\}_{i=1}^{N_k}$. The target domain is $(\boldsymbol{\Sigma}_{\mathcal{T},i})_{i=1}^{N_{\mathcal{T}}}$ with the average outcome $\bar{y}_{\mathcal{T}}$.

## 2 Regression modeling from covariance matrices using Riemannian geometry

**Riemannian geometry of $\mathbb{S}_d^{++}$** The covariance matrices belong to the set of $d \times d$ symmetric positive definite matrices denoted $\mathbb{S}_d^{++}$ [51, 44]. The latter is open in the set of $d \times d$ symmetric matrices denoted $\mathbb{S}_d$, and thus $\mathbb{S}_d^{++}$ is a smooth manifold [9]. A vector space is defined at each $\boldsymbol{\Sigma} \in \mathbb{S}_d^{++}$, called the tangent space, denoted $T_{\boldsymbol{\Sigma}} \mathbb{S}_d^{++}$, and is equal to $\mathbb{S}_d$, the ambient space. Equipped with a smooth inner product at every tangent space, a smooth manifold b ecomes a Riemannian manifold. To do so, we make use of the affine invariant Riemannian metric [51, 44]. Given $\boldsymbol{\Gamma}, \boldsymbol{\Gamma}' \in T_{\boldsymbol{\Sigma}} \mathbb{S}_d^{++}$, this metric is $\langle \boldsymbol{\Gamma}, \boldsymbol{\Gamma}' \rangle_{\boldsymbol{\Sigma}} = \text{tr}\left(\boldsymbol{\Sigma}^{-1} \boldsymbol{\Gamma} \boldsymbol{\Sigma}^{-1} \boldsymbol{\Gamma}'\right)$.

**Riemannian mean** The Riemannian distance (or geodesic distance) associated with the affine invariant metric is $\delta_R(\boldsymbol{\Sigma}, \boldsymbol{\Sigma}') \triangleq \left\|\log\left(\boldsymbol{\Sigma}^{-1/2} \boldsymbol{\Sigma}' \boldsymbol{\Sigma}^{-1/2}\right)\right\|_F$ with $\log : \mathbb{S}_d^{++} \to \mathbb{S}_d$ being the matrix logarithm (see Appendix A.1 for a definition). This distance is used to compute the Riemannian mean $\bar{\boldsymbol{\Sigma}}$ defined for a set $\{\boldsymbol{\Sigma}_i\}_{i=1}^N \subset \mathbb{S}_d^{++}$ as $\bar{\boldsymbol{\Sigma}} \triangleq \arg\min_{\boldsymbol{\Sigma} \in \mathbb{S}_d^{++}} \sum_{i=1}^N \delta_R(\boldsymbol{\Sigma}, \boldsymbol{\Sigma}_i)^2$.

This mean is efficiently computed with a Riemannian gradient descent [44, 63].

**Riemannian logarithmic mapping**  The idea of the covariance-based approach is to define non-linear feature transformations into vectors that can be used as input for classical linear machine learning models. To do so, the Riemannian logarithmic mapping of $\mathbf{\Sigma}'$ at $\mathbf{\Sigma}$ is defined as

$$\log_{\mathbf{\Sigma}}(\mathbf{\Sigma}') \triangleq \mathbf{\Sigma}^{1/2} \log\left(\mathbf{\Sigma}^{-1/2}\mathbf{\Sigma}'\mathbf{\Sigma}^{-1/2}\right)\mathbf{\Sigma}^{1/2} \in T_{\mathbf{\Sigma}}\mathbb{S}_d^{++} \ . \tag{1}$$

Thus, matrices in the Riemannian manifold $\mathbb{S}_d^{++}$ are transformed into tangent vectors.

**Parallel transport**  A classical practice to align distributions is parallel transport of covariance matrices from their mean to the identity and then apply the logarithmic mapping (1). Parallel transport along a curve allows to move SPD matrices from one point on the curve to another point on the curve while keeping the inner product between the logarithmic mappings with any other vector transported along the same curve constant. The following lemma gives the parallel transport of $\mathbf{\Sigma}'$ from $\mathbf{\Sigma}$ to $\mathbf{I}_d$ along the geodesic between these two points (See proof in Appendix A.2).

**Lemma 2.1 (Parallel transport to the identity).** *Given $\mathbf{\Sigma}, \mathbf{\Sigma}' \in \mathbb{S}_d^{++}$, the parallel transport of $\mathbf{\Sigma}'$ along the geodesic from $\mathbf{\Sigma}$ to the identity $\mathbf{I}_d$ at $\alpha \in [0,1]$ is*

$$\mathrm{PT}\left(\mathbf{\Sigma}', \mathbf{\Sigma}, \alpha\right) \triangleq \mathbf{\Sigma}^{-\alpha/2}\mathbf{\Sigma}'\mathbf{\Sigma}^{-\alpha/2} \ .$$

**Learning on $\mathbb{S}_d^{++}$**  The logarithmic mapping (1) at the identity is simply the matrix logarithm. Thus, a classical non-linear feature extraction [6, 36, 8] of a dataset $\{\mathbf{\Sigma}_i\}_{i=1}^N$ of Riemannian mean $\overline{\mathbf{\Sigma}}$ combines parallel transport and logarithmic mapping at the identity,

$$\phi\left(\mathbf{\Sigma}_i, \overline{\mathbf{\Sigma}}\right) \triangleq \mathrm{uvec}\left(\log_{\mathbf{I}_d}\left(\mathrm{PT}\left(\mathbf{\Sigma}_i, \overline{\mathbf{\Sigma}}, 1\right)\right)\right) = \mathrm{uvec}\left(\log\left(\overline{\mathbf{\Sigma}}^{-1/2}\mathbf{\Sigma}_i\overline{\mathbf{\Sigma}}^{-1/2}\right)\right) \in \mathbb{R}^{d(d+1)/2} \tag{2}$$

where uvec vectorizes the upper triangular part with off-diagonal elements multiplied by $\sqrt{2}$ to preserve the norm. Correcting dataset shifts by re-centering all source datasets [62], corresponds to parallel transporting data $\{\mathbf{\Sigma}_{k,i}\}_{i=1}^{N_k}$ of each domain $k \in [\![1, K]\!]$ from its Riemannian mean $\overline{\mathbf{\Sigma}}_k$ to the identity,

$$\phi(\mathbf{\Sigma}_{k,i}, \overline{\mathbf{\Sigma}}_k) = \mathrm{uvec}\left(\log\left(\overline{\mathbf{\Sigma}}_k^{-1/2}\mathbf{\Sigma}_{k,i}\overline{\mathbf{\Sigma}}_k^{-1/2}\right)\right) \ . \tag{3}$$

This method is the go-to approach for reducing shifts of the covariance matrix distributions coming from different domains and has been applied successfully for brain-computer interfaces [45, 59] and age prediction from M/EEG data [36].

## 3  Learning to recenter from highly shifted $y$ distributions with GOPSA

In this section, we develop a novel multi-source domain adaptation method, called Geodesic Optimization for Predictive Shift Adaptation (GOPSA), that operates on the $\mathbb{S}_d^{++}$ manifold and is capable of handling vastly different distributions of $y$. Our approach implements a Riemannian mixed-effects model, which consists of two components: a single parameter estimating a geodesic intercept specific to each domain and a set of parameters shared across domains.

At train-time, GOPSA jointly learns the parallel transport of each of the $K$ source domains and the regression model shared across domains. At test-time, we assume having access to the target mean response value $\bar{y}_{\mathcal{T}}$ and predict on the unlabeled target domain of covariance matrices $(\mathbf{\Sigma}_{\mathcal{T},i})_{i=1}^{N_{\mathcal{T}}}$. GOPSA focuses solely on learning the parallel transport of the target domain so that the mean prediction, using the regression model learned at train-time, matches $\bar{y}_{\mathcal{T}}$.

**Parallel transport along the geodesic**  In Section 2, we presented how domain adaptation is performed on $\mathbb{S}_d^{++}$. In particular, (3) presents how to account for data shifts of each domain. However, this operator can only work if the variability between domains is considered as noise. As explained earlier, we are interested in shifts in both features and the response variable. Thus, (3) discards shift coming from the response variable and hence harms the performance of the predictive model. Based on the Lemma 2.1, we propose to parallel transport features to any point on the geodesic between a domain-specific Riemannian mean $\overline{\mathbf{\Sigma}}_k$ and the identity. Indeed, GOPSA parallel transports $\mathbf{\Sigma}_{k,i}$ on this geodesic with $\alpha \in [0,1]$ and then applies the Riemannian logarithmic mapping (1) at the identity,

$$\phi(\mathbf{\Sigma}_{k,i}, \overline{\mathbf{\Sigma}}_k, \alpha) \triangleq \mathrm{uvec}\left(\log_{\mathbf{I}_d}\left(\mathrm{PT}\left(\mathbf{\Sigma}_{k,i}, \overline{\mathbf{\Sigma}}_k, \alpha\right)\right)\right) = \mathrm{uvec}\left(\log\left(\overline{\mathbf{\Sigma}}_k^{-\alpha/2}\mathbf{\Sigma}_{k,i}\overline{\mathbf{\Sigma}}_k^{-\alpha/2}\right)\right). \tag{4}$$

This allows each domain to undergo parallel transport to a certain degree, effectively moving it toward the identity.

| **Algorithm 1:** Train-Time GOPSA | **Algorithm 2:** Test-Time GOPSA |
|---|---|
| **Input:** For all $k \in [\![1, K]\!]$, $\{(\mathbf{\Sigma}_{k,i}, y_{k,i})\}_{i=1}^{N_k}$, initialization of $\mathbf{\gamma}_{\mathcal{S}}$, step-sizes $\{\xi_t\}_{t \geq 1}$ | **Input:** $\{\mathbf{\Sigma}_{\mathcal{T},i}\}_{i=1}^{N_{\mathcal{T}}}$, mean outcome value $\bar{y}_{\mathcal{T}}$, initialization of $\gamma_{\mathcal{T}}$, trained Ridge coeff. $\mathbf{\beta}_{\mathcal{S}}^{\star}(\gamma_{\mathcal{S}}^{\star})$, step-sizes $\{\xi_t\}_{t \geq 1}$ |

**Algorithm 1:** Train-Time GOPSA

**Input:** For all $k \in [\![1, K]\!]$, $\{(\mathbf{\Sigma}_{k,i}, y_{k,i})\}_{i=1}^{N_k}$,
    initialization of $\mathbf{\gamma}_{\mathcal{S}}$, step-sizes $\{\xi_t\}_{t \geq 1}$
**for** $k = 1 \to K$ **do**
    $\overline{\mathbf{\Sigma}}_k \leftarrow$ Riemannian mean of $\{\mathbf{\Sigma}_{k,i}\}_{i=1}^{N_k}$
**end**
$t \leftarrow 1$
**while** *not converged* **do**
    $\mathbf{Z}_{\mathcal{S}}(\mathbf{\gamma}_{\mathcal{S}}) \leftarrow$ Compute features with (7)
    $\mathbf{\beta}_{\mathcal{S}}^{\star}(\mathbf{\gamma}_{\mathcal{S}}) \leftarrow$ Compute Ridge coeff. with (8)
    $\nabla\mathcal{L}_{\mathcal{S}}(\mathbf{\gamma}_{\mathcal{S}}) \leftarrow$ Compute loss gradient of (8)
    $\mathbf{\gamma}_{\mathcal{S}} \leftarrow \mathbf{\gamma}_{\mathcal{S}} - \xi_t \nabla\mathcal{L}_{\mathcal{S}}(\mathbf{\gamma}_{\mathcal{S}})$
    $t \leftarrow t + 1$
**end**
**return** $\mathbf{\beta}_{\mathcal{S}}^{\star}(\mathbf{\gamma}_{\mathcal{S}}^{\star})$

**Algorithm 2:** Test-Time GOPSA

**Input:** $\{\mathbf{\Sigma}_{\mathcal{T},i}\}_{i=1}^{N_{\mathcal{T}}}$, mean outcome value $\bar{y}_{\mathcal{T}}$,
    initialization of $\gamma_{\mathcal{T}}$, trained Ridge
    coeff. $\mathbf{\beta}_{\mathcal{S}}^{\star}(\gamma_{\mathcal{S}}^{\star})$, step-sizes $\{\xi_t\}_{t \geq 1}$
$\overline{\mathbf{\Sigma}}_{\mathcal{T}} \leftarrow$ Riemannian mean of $\{\mathbf{\Sigma}_{\mathcal{T},i}\}_{i=1}^{N_{\mathcal{T}}}$
$t \leftarrow 1$
**while** *not converged* **do**
    $\mathbf{Z}_{\mathcal{T}}(\gamma_{\mathcal{T}}) \leftarrow$ Compute features with (9)
    $\mathcal{L}_{\mathcal{T}}'(\gamma_{\mathcal{T}}) \leftarrow$ Compute loss derivative
    of (10)
    $\gamma_{\mathcal{T}} \leftarrow \gamma_{\mathcal{T}} - \xi_t \mathcal{L}_{\mathcal{T}}'(\gamma_{\mathcal{T}})$
    $t \leftarrow t + 1$
**end**
$\widehat{\mathbf{y}}_{\mathcal{T}} \leftarrow \mathbf{Z}_{\mathcal{T}}(\gamma_{\mathcal{T}}^{\star})\mathbf{\beta}_{\mathcal{S}}^{\star}(\gamma_{\mathcal{S}}^{\star})$
**return** $\widehat{\mathbf{y}}_{\mathcal{T}}$

### 3.1 Train-time

GOPSA aims to learn simultaneously features from (4) and a regression model. To do so, we solve the following optimization problem

$$\underset{\substack{\mathbf{\beta}_{\mathcal{S}} \in \mathbb{R}^{d(d+1)/2} \\ \mathbf{\alpha}_{\mathcal{S}} \in [0,1]^K}}{\text{minimize}} \sum_{k=1}^{K} \sum_{i=1}^{N_k} \left( y_{k,i} - \mathbf{\beta}_{\mathcal{S}}^{\top} \phi\left(\mathbf{\Sigma}_{k,i}, \overline{\mathbf{\Sigma}}_k, \alpha_k\right)\right)^2 \tag{5}$$

with $\mathbf{\alpha}_{\mathcal{S}} = [\alpha_1, \ldots, \alpha_K]^{\top}$. This cost function is decomposed into three key aspects. First, covariance matrices undergo parallel transported using Lemma 2.1 to account for shifts between domains. Second, they are vectorized, and a linear regression predicts the output variable from these vectorized features. Third, the coefficients of the linear regression $\mathbf{\beta}_{\mathcal{S}}$ and the $\mathbf{\alpha}_{\mathcal{S}}$ are learned jointly so that the predictor is adapted to the parallel transport and reciprocally. Besides, to enforce the constraint on $\mathbf{\alpha}_{\mathcal{S}}$, we re-parameterize it using the sigmoid function, which defines a bijection between $\mathbb{R}$ and $(0, 1)$, thereby ensuring that the resulting $\mathbf{\alpha}_{\mathcal{S}}$ values lie within the desired range: $\alpha_k = \sigma(\gamma_k) \triangleq (1 + \exp(-\gamma_k))^{-1}$. Thus, the constrained problem (5) can be formulated as the following unconstrained optimization problem

$$\underset{\substack{\mathbf{\beta}_{\mathcal{S}} \in \mathbb{R}^{d(d+1)/2} \\ \mathbf{\gamma}_{\mathcal{S}} \in \mathbb{R}^K}}{\text{minimize}} \sum_{k=1}^{K} \sum_{i=1}^{N_k} \left( y_{k,i} - \mathbf{\beta}_{\mathcal{S}}^{\top} \phi\left(\mathbf{\Sigma}_{k,i}, \overline{\mathbf{\Sigma}}_k, \sigma(\gamma_k)\right)\right)^2 , \tag{6}$$

with $\mathbf{\gamma}_{\mathcal{S}} = [\gamma_1, \ldots, \gamma_K]^{\top}$. Let us define the matrix $\mathbf{Z}_{\mathcal{S}}(\mathbf{\gamma}) \in \mathbb{R}^{N_{\mathcal{S}} \times d(d+1)/2}$, with $N_{\mathcal{S}} = \sum_{k=1}^{K} N_k$, as the concatenation of the source data, where each row corresponds to a feature vector:

$$\mathbf{Z}_{\mathcal{S}}(\mathbf{\gamma}) = \left[ \phi\left(\mathbf{\Sigma}_{1,1}, \overline{\mathbf{\Sigma}}_1, \sigma(\gamma_1)\right), \ldots, \phi\left(\mathbf{\Sigma}_{K,N_K}, \overline{\mathbf{\Sigma}}_K, \sigma(\gamma_K)\right)\right]^{\top} . \tag{7}$$

In the same manner, the source labels are concatenated to $\mathbf{y}_{\mathcal{S}} = [y_{1,1}, \ldots, y_{K,N_K}]^{\top} \in \mathbb{R}^{N_{\mathcal{S}}}$. Given a fixed $\mathbf{\gamma}_{\mathcal{S}}$, the problem (6) is solved with the ordinary least squares estimator. In practice, we choose to regularize the estimation of the linear regression with a Ridge penalty. Thus, (6) is rewritten as

$$\mathbf{\gamma}_{\mathcal{S}}^{\star} \triangleq \underset{\mathbf{\gamma} \in \mathbb{R}^K}{\arg\min} \left\{ \mathcal{L}_{\mathcal{S}}(\mathbf{\gamma}) \triangleq \|\mathbf{y}_{\mathcal{S}} - \mathbf{Z}_{\mathcal{S}}(\mathbf{\gamma})\mathbf{\beta}_{\mathcal{S}}^{\star}(\mathbf{\gamma})\|_2^2 \right\}$$

$$\text{subject to} \quad \mathbf{\beta}_{\mathcal{S}}^{\star}(\mathbf{\gamma}) \triangleq \mathbf{Z}_{\mathcal{S}}(\mathbf{\gamma})^{\top}(\lambda \mathbf{I}_N + \mathbf{Z}_{\mathcal{S}}(\mathbf{\gamma})\mathbf{Z}_{\mathcal{S}}(\mathbf{\gamma})^{\top})^{-1}\mathbf{y}_{\mathcal{S}} , \tag{8}$$

where $\mathbf{\beta}_{\mathcal{S}}^{\star}(\mathbf{\gamma}) \in \mathbb{R}^{d(d+1)/2}$ are the Ridge estimated coefficients given a fixed $\mathbf{\gamma}$ and $\lambda > 0$ is the regularization hyperparameter. The problem (8) is efficiently solved with any gradient-based solver [39].

The train-time of GOPSA is summarized in Algorithm 1. The proposed training algorithm begins by calculating the Riemannian mean of covariance matrices for each domain $k$. It then iteratively optimizes the parameters $\mathbf{\gamma}_{\mathcal{S}}$ by computing the feature matrix (7), determining Ridge regression coefficients (8), and updating $\mathbf{\gamma}_{\mathcal{S}}$ using a gradient descent step on the loss function (8) until convergence. The output result is the optimized Ridge regression coefficients. For clarity of presentation,

Algorithm 1 employs a gradient descent. In practice, we use L-BFGS and obtain the gradient using automatic differentiation through the Ridge solution that is plugged into the loss in (8).

## 3.2 Test-time

At test-time, we now have a fitted linear model on source data with coefficients $\boldsymbol{\beta}_{\mathcal{S}}^{\star}(\gamma_{\mathcal{S}}^{\star})$. The goal is to adapt a new target domain $(\boldsymbol{\Sigma}_{\mathcal{T},i})_{i=1}^{N_{\mathcal{T}}}$ for which the average outcome $\bar{y}_{\mathcal{T}}$ is assumed to be known. First, let us define the matrix $\boldsymbol{Z}_{\mathcal{T}}(\gamma) \in \mathbb{R}^{N_{\mathcal{T}} \times d(d+1)/2}$ as the concatenation of the target data

$$\boldsymbol{Z}_{\mathcal{T}}(\gamma) = \left[ \phi\left(\boldsymbol{\Sigma}_{\mathcal{T},1}, \overline{\boldsymbol{\Sigma}}_{\mathcal{T}}, \sigma(\gamma)\right), \ldots, \phi\left(\boldsymbol{\Sigma}_{\mathcal{T},N_{\mathcal{T}}}, \overline{\boldsymbol{\Sigma}}_{\mathcal{T}}, \sigma(\gamma)\right) \right]^{\top} . \tag{9}$$

Then, GOPSA adapts to this new target domain by minimizing the error between $\bar{y}_{\mathcal{T}}$ and its estimation computed with the fitted linear model. This minimization is performed with respect to $\gamma_{\mathcal{T}} \in \mathbb{R}$ that parametrizes the parallel transport of the target domain, i.e.,

$$\gamma_{\mathcal{T}}^{\star} = \arg\min_{\gamma \in \mathbb{R}} \left\{ \mathcal{L}_{\mathcal{T}}(\gamma) \triangleq \left( \bar{y}_{\mathcal{T}} - \frac{1}{N_{\mathcal{T}}} \mathbf{1}_{N_{\mathcal{T}}}^{\top} \boldsymbol{Z}_{\mathcal{T}}(\gamma) \boldsymbol{\beta}_{\mathcal{S}}^{\star}(\gamma_{\mathcal{S}}^{\star}) \right)^{2} \right\} . \tag{10}$$

Finally, the predictions on the target domain are

$$\widehat{\boldsymbol{y}}_{\mathcal{T}} = \boldsymbol{Z}_{\mathcal{T}}(\gamma_{\mathcal{T}}^{\star}) \boldsymbol{\beta}_{\mathcal{S}}^{\star}(\gamma_{\mathcal{S}}^{\star}) \in \mathbb{R}^{N_{\mathcal{T}}} . \tag{11}$$

The test-time procedure of GOPSA is summarized in Algorithm 2. The algorithm begins by calculating the Riemannian mean of the target covariance matrices $\{\boldsymbol{\Sigma}_{\mathcal{T},i}\}_{i=1}^{N_{\mathcal{T}}}$. It then iteratively optimizes the parameter $\gamma_{\mathcal{T}}$ by computing the feature matrix (9), the derivative of the loss function (10), and updating $\gamma_{\mathcal{T}}$ using a gradient descent step until convergence. The algorithm determines the estimated target outcomes, $\widehat{\boldsymbol{y}}_{\mathcal{T}}$, by using the optimized $\gamma_{\mathcal{T}}^{\star}$ on the feature matrix, combined with the pre-trained regression coefficients $\boldsymbol{\beta}_{\mathcal{S}}^{\star}(\gamma_{\mathcal{S}}^{\star})$. The output result is the predicted target outcomes $\widehat{\boldsymbol{y}}_{\mathcal{T}}$. It should be noted that, once again, for clarity of presentation, Algorithm 2 employs a gradient descent, but other derivative-based optimization methods can be used.

# 4 Empirical benchmarks

In this section, we built empirical benchmark to evaluate the performance of GOPSA. We first present the simulated data that we used to illustrate the relevance of our method when there is a joint distribution shift of the data and the labels. Then, we present the EEG dataset that we used to evaluate the performance of GOPSA with real data from different recording sites. Finally, we present the baseline methods that are compared with GOPSA.

**Simulated data** To generate simulated data, we used the generative model described in [47, 48, 36]. The data follow the classical instantaneous mixing model:

$$\boldsymbol{x}_i(t) = \boldsymbol{A}\boldsymbol{s}_i(t) \tag{12}$$

where $\boldsymbol{x}_i(t) \in \mathbb{R}^d$ are the observed time-series, $\boldsymbol{s}_i(t) \in \mathbb{R}^d$ are the underlying signal of the neural generators and $\boldsymbol{A}$ is the mixing matrix whose columns are the observed spatial patterns of the neural generators. Furthermore, we assume that $y$ follow a log-linear model:

$$y_i = \beta_0 + \sum_{\ell=1}^{d} \beta_\ell \log(p_{\ell i}) \tag{13}$$

where $p_{\ell i} > 0$ is the variance of the $\ell$-th element of the underlying signal $\boldsymbol{s}_i(t)$ as introduced in [47, 48, 36]. From this, we generate domains (source and target) by applying shifts on $X$ and $y$. To do so, we introduced a per-domain shift in the data distribution by applying an affine transformation to the covariance matrices $\boldsymbol{\Sigma}_i \triangleq \mathbb{E}[\boldsymbol{x}_i(t)\boldsymbol{x}_i(t)^{\top}]$:

$$\boldsymbol{\Sigma}_i \mapsto \boldsymbol{B}_k^{\xi} \boldsymbol{\Sigma}_i \boldsymbol{B}_k^{\xi} \tag{14}$$

with $\boldsymbol{B}_k \in \mathbb{S}_d^{++}$ and $\xi \geq 0$ controlling the amplitude of the shift. Then, we shifted the label distribution by modifying the variance of the underlying signal $p_{\ell i}$:

$$p_{\ell i} \mapsto p_{\ell i}^{1+k\xi} \tag{15}$$

with $\xi \geq 0$ still controlling the amplitude of the shift. Thus, the distribution of $y$ is shifted per domain because of the log-linear relationship of (13). It should be noted that $\boldsymbol{\beta}$ is kept constant across domains.

**HarMNqEEG dataset** The HarMNqEEG dataset [33] was used for our numerical experiments. This dataset includes EEG recordings collected from 1564 participants across 14 different study sites, distributed across 9 countries. In our analysis, we consider each study site as a distinct domain. Appendix A.5 provides detailed demographic information. The EEG data were recorded with the same montage of 19 channels of the 10/20 International Electrodes Positioning System. The dataset provides pre-computed cross-spectral tensors for each participant rather than raw data, and anonymized metadata including the age and the sex of the participants. More precisely, the shared data consists of cross-spectral matrices with a frequency range of $1.17\,\mathrm{Hz}$ to $19.14\,\mathrm{Hz}$, sampled at a resolution of $0.39\,\mathrm{Hz}$. A standardized recording protocol was enforced to ensure the consistency across EEG recording of the dataset. In addition to recording constraints, this protocol included artifact cleaning procedures. The cross-spectrum were computed using Bartlett's method (See Appendix A.3). Our pre-processing steps were guided by the pre-processing pipeline outlined in [33]. First, we performed a common average reference (CAR) on all cross-spectrum (See Appendix A.3) as different EEG references were used across domains. Subsequently, we extracted the real part of the cross-spectral tensor to obtain co-spectrum tensors containing frequency-specific covariance estimates along the frequency spectrum. Due to the linear dependence between channels introduced by the CAR, the covariance matrices are rank deficient. To address this, we applied a shrinkage regularization with a coefficient of $10^{-5}$ to the data. Additionally, we implemented a global-scale factor (GSF) correction, which compensates for amplitude variations between EEG recordings by scaling the covariance matrices with a subject-specific factor [24, 33] (See Appendix A.3). Following these pre-processing steps, we obtained a set of 49 covariance matrices for each EEG recording, with each matrix corresponding to a specific frequency bin of the EEG signal. This pre-processed co-spectrum served as the input data for our domain adaptation study.

**Performance evaluation and hyperparameter selection** To evaluate the performance of the compared methods, we conducted experiments across several combinations of source and target sites. We selected source domains such that the union distribution of their predictive variable $y$ encompasses a broad age range. All remaining sites were assigned as target domains. For each source-target combination we performed a stratified shuffle split approach with 100 repetitions on the target data. Stratification was based on the recording sites to ensure that each split contained a balanced proportion of participants from each site. The regularization parameter $\lambda$ in Ridge regression was selected with a nested cross-validation (grid search) over a logarithmic grid of values from $10^{-1}$ to $10^5$. To evaluate the benefit of `GOPSA`, we compared it against four baselines. Detailed mathematical formulations of these baselines can be found in Appendix A.4. For each baseline method, the regression task was performed with Ridge regression.

**Domain-aware dummy model** (`DO Dummy`) As `GOPSA` requires access to the mean $\bar{y}_k$ of each domain, we used a domain-aware dummy model predicting always the mean $\bar{y}_k$ of each domain.

**No re-center / No domain adaptation** (`No DA`) This second baseline method involves applying the regression pipeline outlined in [47, 48] without any re-centering. In this setup, all covariance matrices are projected to the tangent space at the source geometric mean $\overline{\boldsymbol{\Sigma}}$ computed from all source points, no matter their recording sites.

**Re-center to a common reference point** (`Re-center` **and** `Re-scale`) As introduced in Section 2, a common transfer learning approach is a Riemannian re-centering of all domains to a common point on the manifold [62, 36]. This baseline thus correspond to re-centering each domain $k$, source and target, independently by whitening them by their respective geometric mean $\overline{\boldsymbol{\Sigma}}_k$. An extension of this approach is to perform a Riemannian re-scaling of all domains to a common dispersion, as presented in [45, 36].

**Domain-aware intercept** (`DO Intercept`) This method consists in fitting one intercept $\beta_0$ per domain. In practice since we assume to know $\bar{y}_\mathcal{T}$, we correct the predicted values so that their mean is equal to $\bar{y}_\mathcal{T}$. This approach is in line with defining mixed-effects models on the Riemannian manifold [33].

**Deep learning** (`GREEN`) The `GREEN` model [40] is a deep-learning architecture tailored for EEG applications like age prediction. Since the HarMNqEEG dataset consists of covariance matrices, we

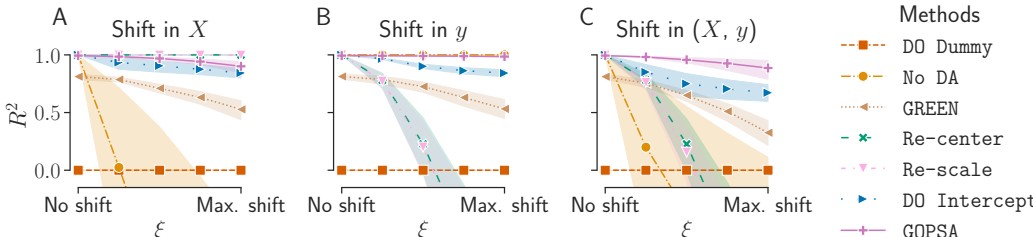

Figure 2: $R^2$ **scores ↑ for different methods on simulated data.** Performance is measured across 5 source domains and 1 target domain, with shifts controlled by $\xi$ (0 to maximum). Data are generated 100 times, with 5 sensors and 300 covariance matrices per domain. The target domain is randomly selected between the 6 domains generated as presented in Section 4, with the remaining domains used as sources. **(A)** A shift is applied on the covariance matrices following (14). **(B)** A shift is applied on the variances following (15). **(C)** Both shifts from (14) and (15) are applied simultaneously.

used the 'G2' variant of GREEN, which starts at the covariance matrices level and includes pooling layers. This variant is designed for SPD matrices, making it an SPD network [26]. Although GREEN has been evaluated on multiple datasets for various predictive tasks, it has not yet been applied in a domain adaptation context and does not include an adaptation layer.

We applied the domain-adaptation methods independently to each of the 49 frequency bins, resulting in 49 geometric means per domain, except for GREEN, which processes all frequency bands simultaneously. $\bar{y}_{\mathcal{T}}$ of each domain was estimated on target splits (50% of the data) that do not overlap with the evaluation target splits (50% remaining). Statistical inference for model comparisons was implemented with a corrected t-test following [37]. Experiments with 100 repetitions and all site combinations have been run on a standard Slurm cluster for 12 hours with 250 CPU cores.

## 5 Results

**Simulated data**  Figure 2 presents the results of simulated experiments where shifts are applied on either $X$, $y$, or both $(X, y)$ as presented in Section 4. All methods were evaluated in three simulation scenarios: shift in $X$ only, shift in $y$ only, and joint shift in $X$ and $y$. The intensity of the shift was controlled by $\xi$ in all scenarios. If there is no shift in $X$, we observe that No DA perfectly estimates the $y$ because the log-linear model is easily estimated across domains even when the $y$ distribution changes (Figure 2 B). The performance of No DA however drops when a shift in $X$ is introduced (Figure 2 A and C). Re-center and Re-scale led to the same results as no scaling shift was applied in the simulation. Both were able to correct the shift in $X$, but performed poorly when a shift in $y$ was added (Figure 2 B and C). GREEN notably showed consistant performance across all scenarios, and was relatively resistant to both types of shifts given it is not designed for domain adaptation. DO Intercept and GOPSA showed the best performance across all scenarios, with an advantage for GOPSA. The interest of GOPSA is to estimate this log-linear model with shifts in $(X, y)$ per domain (Figure 2 C) which other methods were not able to do. These experiments demonstrate the efficiency of the proposed method in estimating shifts in $X$ between domains even in the presence of a shift in $y$, contrary to the baseline methods. Theoretically, based on the generative model of the simulated data, the data $X$ and outcome $y$ are linked by a log-linear relationship. This implies that, knowing the shift in $X$ for the target domain, predictions can be made even when $y$ distributions do not overlap between the source and target. Since GOPSA estimates the target shift in $X$ by minimizing $(\bar{y} - \text{mean}(\hat{y}_i))^2$, it is capable of handling such scenarios.

**HarMNqEEG data**  We computed benchmarks for five combinations of source sites and we displayed the results for the three metrics selected for performance evaluation, each colored box representing one method (Figure 3). A min-max normalization was applied to each site combinations separately across methods. We first conducted model comparisons in terms of absolute performance across all baselines (**A**). No DA, without domain specific re-centering, performed worse than DO Dummy in terms of $R^2$ score and MAE. Re-center and Re-scale led to lower performances across all metrics, which can be expected as the Riemannian mean is correlated with age in our problem setting Figure 1. Eventhough its architecture does not include an adaptation layer, GREEN

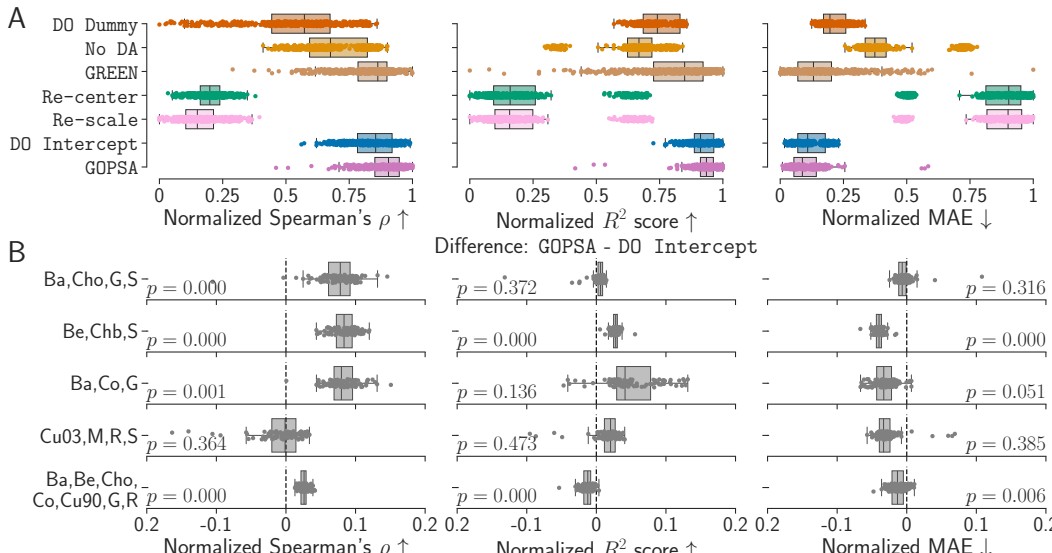

Figure 3: **Normalized performance of the different methods on several source-target combinations for three metrics:** Spearman's $\rho \uparrow$ (left), $R^2$ score $\uparrow$ (middle) and Mean Absolute Error $\downarrow$ (right). As a large variability in the score values was present between the site combinations, we applied a min-max normalization per combination to set the minimum score across all methods to 0 and the maximum score to 1. (**A**) Boxplot of the concatenated results for the three normalized scores. One point corresponds to one split of one site combination. (**B**) Boxplots of the difference between the normalized scores of GOPSA and DO Intercept. A row corresponds to one site combination, one point corresponds to one split. For each plot, the associated results of Nadeau's & Bengio's corrected t-test [37] are displayed. A p-value lower than 0.05 indicates a significant difference between the two methods. Ba: Barbados, Be: Bern, Chb: CHBMP (Cuba), Co: Columbia, Cho: Chongqing, Cu03: Cuba2003, Cu90: Cuba90, G: Germany, M: Malaysia, R: Russia, S: Switzerland

reached better performance than the previous methods mentionned, but lacked consistency across site combinations and metrics with large variance especially for the $R^2$ score and MAE. For all scores, DO Intercept and GOPSA reached the best average performance with lower variance. A version of Figure 3 **A** without normalization is presented in Appendix A.7. As DO intercept and GOPSA showed overlapping performance distributions, we investigated their paired split-wise (non-rescaled) score differences (**B**). The site-specific differences of GOPSA scores minus DO Intercept are displayed with their associated p-values. For one site combination (Ba,Be,Cho,Co,Cu90,G,R), DO Intercept yielded higher $R^2$ scores, and no significant difference was found between the two methods for Ba,Co,G. Similarly, no significant difference was observed on Spearman's $\rho$ results for Cu03,M,R,S. Overall, GOPSA significantly outperformed DO Intercept in five site combinations for MAE, four for Spearman's $\rho$ and three for $R^2$ score. Detailed results for each source-target combination are presented in Appendix A.6 for Spearman's $\rho$, $R^2$ score, and MAE. The bottom rows correspond to the mean performance of each method of all site combinations, and their average standard deviation (see Appendix A.8 for associated boxplots). We expected GOPSA to outperform the baseline methods (e.g. DO Intercept) whenever joint $(X, y)$ shifts occur. In our experimental benchmark, GOPSA significantly outperformed the baseline methods in some site combinations, but not all. This allows us to assume that not all site combinations show joint shifts.

**Model inspection** Next, we investigated the impact of the different re-centering approaches on the data Figure 4. Power spectrum densities (PSDs) were computed as the mean across sensors of the diagonals of the covariance matrices Riemannian mean for each site combination after No DA, Re-center and GOPSA (**A**). PSDs for No DA display the initial variability between sites without recentering (cf. Figure 1). Re-center resulted in flat PSDs because all data were re-centered to the identity. PSDs produced by GOPSA are flattened and more similar across sites compared to No DA without removing too much information, unlike the un-effective Re-center method (cf. Figure 3). The alpha values are inspected as a function of the site mean age (**B**). Re-center leads to alpha values all equal to one as all sites are re-centered to the identity. For GOPSA, we observed a linear

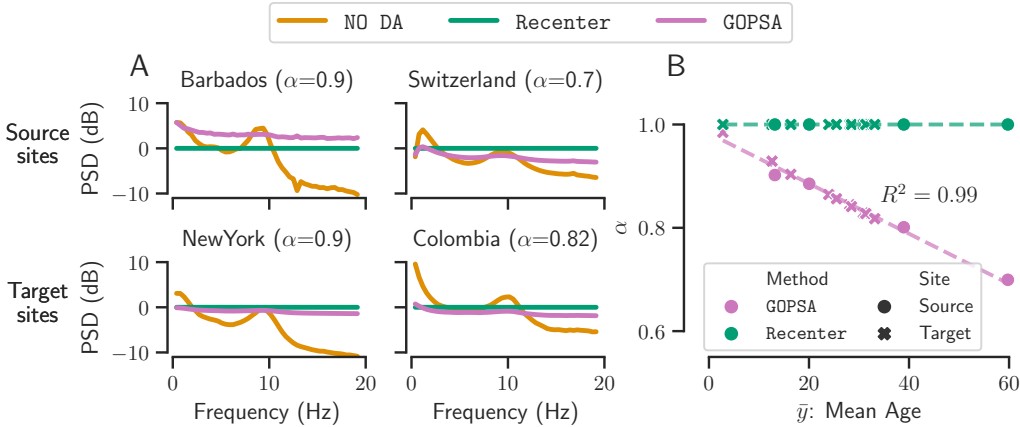

Figure 4: **Model inspection of `GOPSA` versus `No DA` and `Re-center`.** Power Spectral Densities (PSDs) and $\alpha$ values were computed on the source sites Barbados, Chongqing, Germany, and Switzerland. The remaining sites were used as target domains. (**A**) Mean PSDs computed across sensors for `No DA`, `Recenter` and `GOPSA` on two source (Barbados and Switzerland) and two target (New York and Columbia) sites. (**B**) $\alpha$ values versus site's mean age for `Re-center` and `GOPSA`. One point corresponds to one site. The coefficient of determination is reported for the `GOPSA` method.

relationship between alpha and the sites' mean age ($R^2 = 0.99$). This is a direct consequence of the optimization process in `GOPSA`, which thus can be regarded a geodesic intercept in a mixed-effects model. Overall, `GOPSA` effectively re-centered sites with younger participants closer to the identity matrix. Re-centering sites around a common point helped reduce the shift in $X$, while not placing all sites at the exact same reference point helped manage the shift in $y$, hence preserving the statistical associations between $X$ and $y$.

## 6 Conclusion

We proposed a novel multi-source domain adaptation approach that adapts shifts in $X$ and $y$ simultaneously by learning jointly a domain specific re-centering operator and the regression model. `GOPSA` was specifically developed to handle joint shifts in the data distribution and the outcome distribution, as illustrated by the simulations in Figure 2. `GOPSA` is a test-time method that does not require to retrain a model when a new domain is presented. `GOPSA` achieved state-of-the-art performance on the HarMNqEEG [33] dataset with EEG from 14 recording sites and over 1500 participants. Our benchmarks showed a significant gain in performance for three different metrics in a majority of site combinations compared to baseline methods. `GOPSA` can thus be used by researchers as a decision rule to infer the presence of joint shifts and, hence, serve as a tool for data exploration and model interpretation. While we focused on shallow regression models, the implementation of `GOPSA` using PyTorch readily supports its inclusion in more complex Riemannian deep learning models [26, 56, 11, 40, 31]. This direction seems promising given our observation that GREEN – a simple deep net combining Riemannian computation with a fully connected layer - already possessed some intrinsic robustness to data shifts. This may point at the capacity of the fully-connected layer to provide additional non-linear transformations that can accommodate the data-generating scenario in which continuous log-linear generators are modified in a discrete manner by site factors. More generally, it emphasizes the potential of complex nonlinear methods for domain adaptation, in line with a recent study on the same dataset reporting positive generalization results using a kernel method [28]. Furthermore, although this work specifically addresses age prediction, the methodology is applicable to a broader range of regression analyses. While `GOPSA` necessitates knowledge or estimability of the average $\bar{y}$ per domain, this requirement aligns with that of mixed-effects models [16, 25, 61], which are extensively employed in biomedical statistics. By combining mixed-effects modeling with Riemannian geometry for EEG, `GOPSA` opens up various applications at the interface between machine learning and biostatistics, such as, biomarker exploration in large multicenter clinical trials [46, 52, 53].

## Acknowledgment

This work was supported by grants ANR-20-CHIA-0016 and ANR-20-IADJ-0002 to AG while at Inria, ANR-20-THIA-0013 to AM, ANR-22-CE33-0015-01 and ANR-17-CONV-0003 operated by LISN to SC and ANR-22-PESN-0012 to AC under the France 2030 program, all managed by the Agence Nationale de la Recherche (ANR) Competing interests: DE is a full-time employee of F. Hoffmann-La Roche Ltd.

Numerical computation was enabled by the scientific Python ecosystem: Matplotlib [27], Scikit-learn [42], Numpy [21], Scipy [54], PyTorch [41] PyRiemann [3], MNE [19] and SKADA [18].

This work was conducted at Inria, AG is presently employed by Meta Platforms. All the datasets used for this work were accessed and processed on the Inria compute infrastructure.

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

# A  Appendix

## A.1  Matrix operations

Given $\boldsymbol{\Sigma} \in \mathbb{S}_d^{++}$ and its Singular Value Decomposition (SVD) $\boldsymbol{\Sigma} = \boldsymbol{U}\operatorname{diag}(\lambda_1, \ldots, \lambda_d)\boldsymbol{U}^\top$, the matrix logarithm of $\boldsymbol{\Sigma}$ is

$$\log(\boldsymbol{\Sigma}) = \boldsymbol{U}\operatorname{diag}(\log(\lambda_1), \ldots, \log(\lambda_d))\boldsymbol{U}^\top. \tag{16}$$

$\boldsymbol{\Sigma}$ to the power $\alpha \in \mathbb{R}$ is

$$\boldsymbol{\Sigma}^\alpha = \boldsymbol{U}\operatorname{diag}(\lambda_1^\alpha, \ldots, \lambda_d^\alpha)\boldsymbol{U}^\top. \tag{17}$$

## A.2  Proof of Lemma 2.1

First, we recall that the geodesic associated with the affine invariant metric from $\boldsymbol{\Sigma}$ to $\boldsymbol{\Sigma}'$ is

$$\boldsymbol{\Sigma}\sharp_\alpha\boldsymbol{\Sigma}' \triangleq \boldsymbol{\Sigma}^{1/2}\left(\boldsymbol{\Sigma}^{-1/2}\boldsymbol{\Sigma}'\boldsymbol{\Sigma}^{-1/2}\right)^\alpha\boldsymbol{\Sigma}^{1/2} \quad \text{for } \alpha \in [0,1]. \tag{18}$$

Hence, $\boldsymbol{\Sigma}\sharp_\alpha\boldsymbol{I}_d = \boldsymbol{\Sigma}^{1-\alpha}$. From [59], the parallel transport of $\boldsymbol{\Sigma}'$ from $\boldsymbol{\Sigma}_1$ to $\boldsymbol{\Sigma}_2$ is

$$\boldsymbol{E}\boldsymbol{\Sigma}'\boldsymbol{E}^\top \quad \text{with} \quad \boldsymbol{E} \triangleq \boldsymbol{\Sigma}_1^{1/2}\left(\boldsymbol{\Sigma}_1^{-1/2}\boldsymbol{\Sigma}_2\boldsymbol{\Sigma}_1^{-1/2}\right)^{1/2}\boldsymbol{\Sigma}_1^{-1/2}. \tag{19}$$

Hence, the parallel transport of $\boldsymbol{\Sigma}'$ from $\boldsymbol{\Sigma}$ to $\boldsymbol{\Sigma}\sharp_\alpha\boldsymbol{I}_d$ is $\boldsymbol{E}\boldsymbol{\Sigma}'\boldsymbol{E}^\top$ with

$$\begin{aligned} \boldsymbol{E} &\triangleq \boldsymbol{\Sigma}^{1/2}\left(\boldsymbol{\Sigma}^{-1/2}\boldsymbol{\Sigma}^{1-\alpha}\boldsymbol{\Sigma}^{-1/2}\right)^{1/2}\boldsymbol{\Sigma}^{-1/2} \\ &= \boldsymbol{\Sigma}^{1/2}\boldsymbol{\Sigma}^{-\alpha/2}\boldsymbol{\Sigma}^{-1/2} = \boldsymbol{\Sigma}^{-\alpha/2} \end{aligned} \tag{20}$$

which concludes the proof.

## A.3  Cross-spectrum computation and preprocessing

**Bartlett estimator**  From [33], the features provided in the HarMNqEEG dataset have been computed using the Bartlett's estimator by averaging more than 20 consecutive and non-overlapping segments. Thus, data consist of cross-spectral matrices with a frequency range of $f_{\min} = 1.17\,\text{Hz}$ to $f_{\max} = 19.14\,\text{Hz}$, sampled at a resolution of $\Delta\omega = 0.39\,\text{Hz}$. These cross-spectral matrices are denoted $\mathbf{S}_{k,i}(\omega) \in \mathcal{H}_d^{++}$ where $k$ is the site, $i$ the participant and $\omega \in \{f_{\min}, f_{\min} + \Delta\omega, \ldots, f_{\max}\}$.

**Common average reference (CAR)**  The cross-spectrum matrices $\mathbf{S}_i(\omega)$ were re-referenced from their original montages with a CAR:

$$\widetilde{\mathbf{S}}_{k,i}(\omega) \triangleq \mathbf{H}\mathbf{S}_{k,i}(\omega)\mathbf{H}^\top \tag{21}$$

where $\mathbf{H} \triangleq \boldsymbol{I}_d - \mathbf{1}_d\mathbf{1}_d^\top/d$.

**Global Scale Factor (GSF)**  Co-spectrum matrices were re-scaled with an individual scalar $\widehat{\zeta}_{k,i}$ that is calculated as the geometric mean of their power spectrum across sensors and frequencies:

$$\widehat{\zeta}_{k,i} \triangleq \exp\left(\frac{1}{N_\omega d}\sum_{\ell=0}^{N_\omega-1}\sum_{c=1}^d \log\left(\left(\widehat{\mathbf{S}}_{k,i}(f_{\min} + \ell\Delta\omega)\right)_{c,c}\right)\right) \tag{22}$$

where $N_\omega \triangleq \frac{f_{\max} - f_{\min}}{\Delta\omega} + 1$. The GSF correction is then applied to the co-spectrum (the real part of the cross-spectrum) for all frequencies $\omega$:

$$\boldsymbol{\Sigma}_{k,i}(\omega) \triangleq \Re\left(\widetilde{\mathbf{S}}_{k,i}(\omega)\right)/\widehat{\zeta}_{k,i}. \tag{23}$$

The $\boldsymbol{\Sigma}_{k,i}(\omega) \in \mathbb{S}_d^{++}$ are the features used the Section 4.

## A.4 Baselines

The four baseline methods used in this work are detailed in the following. For every methods we have access to $K$ labeled source domains, each including $N_k$ covariance matrices and their corresponding variables of interest $(\mathbf{\Sigma}_{k,i}, y_{k,i})_{i=1}^{N_k}$. The method `DO Dummy` and the mixed-effects model baseline `DO Intercept` both access the mean value $\bar{y}_{\mathcal{T}}$ of the target domain variable to predict. We remind that as the dataset used in the empirical benchmarks is constituted of several frequency bands, each method is applied to each frequency band independently and then computed vectors are concatenated.

**Domain-aware dummy model** (`DO Dummy`) The `DO Dummy` simply returns the mean value $\bar{y}_{\mathcal{T}}$ for every predictions of a given target domain.

**No re-center / No domain adaptation** (`No DA`) The covariance matrices are used as input of the regression pipeline without any re-centering. First, the geometric mean of the source matrices is computed:

$$\overline{\mathbf{\Sigma}}_{\mathcal{S}} \triangleq \underset{\mathbf{\Sigma} \in \mathbb{S}_d^{++}}{\arg\min} \sum_{k=1}^{K} \sum_{i=1}^{N_k} \delta_R(\mathbf{\Sigma}, \mathbf{\Sigma}_{k,i})^2. \tag{24}$$

Then, source feature vectors are extracted with:

$$\phi(\mathbf{\Sigma}_{k,i}, \overline{\mathbf{\Sigma}}_{\mathcal{S}}) = \mathrm{uvec}\left(\log\left(\overline{\mathbf{\Sigma}}_{\mathcal{S}}^{-1/2} \mathbf{\Sigma}_{k,i} \overline{\mathbf{\Sigma}}_{\mathcal{S}}^{-1/2}\right)\right) \in \mathbb{R}^{d(d+1)/2}. \tag{25}$$

Finally, the target feature vectors are extracted from the target data $(\mathbf{\Sigma}_{\mathcal{T},i})_{i=1}^{N_{\mathcal{T}}}$ with:

$$\phi(\mathbf{\Sigma}_{\mathcal{T},i}, \overline{\mathbf{\Sigma}}_{\mathcal{S}}) = \mathrm{uvec}\left(\log\left(\overline{\mathbf{\Sigma}}_{\mathcal{S}}^{-1/2} \mathbf{\Sigma}_{\mathcal{T},i} \overline{\mathbf{\Sigma}}_{\mathcal{S}}^{-1/2}\right)\right) \in \mathbb{R}^{d(d+1)/2}. \tag{26}$$

**Re-center to a common reference point** (`Re-center`) Domains are re-centered to a common reference point, here we decided to use the identity. First, the geometric mean of each domain is computed:

$$\overline{\mathbf{\Sigma}}_k \triangleq \underset{\mathbf{\Sigma} \in \mathbb{S}_d^{++}}{\arg\min} \sum_{i=1}^{N_k} \delta_R(\mathbf{\Sigma}, \mathbf{\Sigma}_{k,i})^2 \ . \tag{27}$$

Then, feature vectors are extracted using the specific geometric mean of each domain:

$$\phi(\mathbf{\Sigma}_{k,i}, \overline{\mathbf{\Sigma}}_k) = \mathrm{uvec}\left(\log\left(\overline{\mathbf{\Sigma}}_k^{-1/2} \mathbf{\Sigma}_{k,i} \overline{\mathbf{\Sigma}}_k^{-1/2}\right)\right) \in \mathbb{R}^{d(d+1)/2} \ . \tag{28}$$

Covariance matrices of the target domain $(\mathbf{\Sigma}_{\mathcal{T},i})_{i=1}^{N_{\mathcal{T}}}$ are also re-centered to the identity using their geometric mean :

$$\overline{\mathbf{\Sigma}}_{\mathcal{T}} \triangleq \underset{\mathbf{\Sigma} \in \mathbb{S}_d^{++}}{\arg\min} \sum_{i=1}^{M} \delta_R(\mathbf{\Sigma}, \mathbf{\Sigma}_{\mathcal{T},i})^2 \tag{29}$$

$$\phi(\mathbf{\Sigma}_{\mathcal{T},i}, \overline{\mathbf{\Sigma}}_{\mathcal{T}}) = \mathrm{uvec}\left(\log\left(\overline{\mathbf{\Sigma}}_{\mathcal{T}}^{-1/2} \mathbf{\Sigma}_i \overline{\mathbf{\Sigma}}_{\mathcal{T}}^{-1/2}\right)\right) \in \mathbb{R}^{d(d+1)/2}. \tag{30}$$

For both `No DA` and `Re-center`, the regression task was performed using a Ridge regression, which included an intercept:

$$\boldsymbol{\beta}_{\mathcal{S}}^{\star}, \beta_{0,\mathcal{S}}^{\star} = \underset{\substack{\boldsymbol{\beta} \in \mathbb{R}^{d(d+1)/2} \\ \beta_0 \in \mathbb{R}}}{\arg\min} \sum_{k=1}^{K} \sum_{i=1}^{N_k} \left(y_{k,i} - \boldsymbol{\beta}^{\top} \boldsymbol{z}_{k,i} - \beta_0\right)^2 + \lambda \|\boldsymbol{\beta}\|_2^2 \tag{31}$$

where $\boldsymbol{z}_{k,i}$ is computed with (25) or (28). The predicted values were computed as:

$$\widehat{y}_{\mathcal{T},i} = (\boldsymbol{\beta}_{\mathcal{S}}^{\star})^{\top} \boldsymbol{z}_{\mathcal{T},i} + \beta_{0,\mathcal{S}}^{\star} \tag{32}$$

where $\boldsymbol{z}_{\mathcal{T},i}$ is computed with (26) or (30).

**Domain-aware intercept** (`DO Intercept`) In addition to the $K$ labeled source domains, we assume to have access to the mean of the variable to predict of the target domain $\bar{y}_{\mathcal{T}}$. At train-time, we fit a Ridge regression with a specific intercept for each domain

$$\boldsymbol{\beta}_{\mathcal{S}}^{\star} = \underset{\boldsymbol{\beta} \in \mathbb{R}^{d(d+1)/2}}{\arg\min} \sum_{k=1}^{K} \sum_{i=1}^{N_k} \left(y_{k,i} - \boldsymbol{\beta}^{\top} \phi(\mathbf{\Sigma}_{k,i}, \overline{\mathbf{\Sigma}}_{\mathcal{S}}) - \bar{y}_k\right)^2 + \lambda \|\boldsymbol{\beta}\|_2^2 \ . \tag{33}$$

Then, at test-time, we fit a new intercept $\beta_{0,\mathcal{T}}$ using the target features:

$$\phi\left(\mathbf{\Sigma}_{\mathcal{T},i}, \overline{\mathbf{\Sigma}}_{\mathcal{S}}\right) = \mathrm{uvec}\left(\log\left(\overline{\mathbf{\Sigma}}_{\mathcal{S}}^{-1/2}\mathbf{\Sigma}_{\mathcal{T},i}\overline{\mathbf{\Sigma}}_{\mathcal{S}}^{-1/2}\right)\right) \in \mathbb{R}^{d(d+1)/2}. \tag{34}$$

The fitted intercept is

$$\beta_{0,\mathcal{T}} = \bar{y}_{\mathcal{T}} - \frac{1}{N_{\mathcal{T}}}\sum_{i=1}^{N_{\mathcal{T}}}(\boldsymbol{\beta}_{\mathcal{S}}^{\star})^{\top}\phi\left(\mathbf{\Sigma}_{\mathcal{T},i}, \overline{\mathbf{\Sigma}}_{\mathcal{S}}\right) \tag{35}$$

and the predictions are

$$\widehat{y}_{\mathcal{T},i} = (\boldsymbol{\beta}_{\mathcal{S}}^{\star})^{\top}\phi\left(\mathbf{\Sigma}_{\mathcal{T},i}, \overline{\mathbf{\Sigma}}_{\mathcal{S}}\right) + \beta_{0,\mathcal{T}} . \tag{36}$$

## A.5 HarMNqEEG dataset

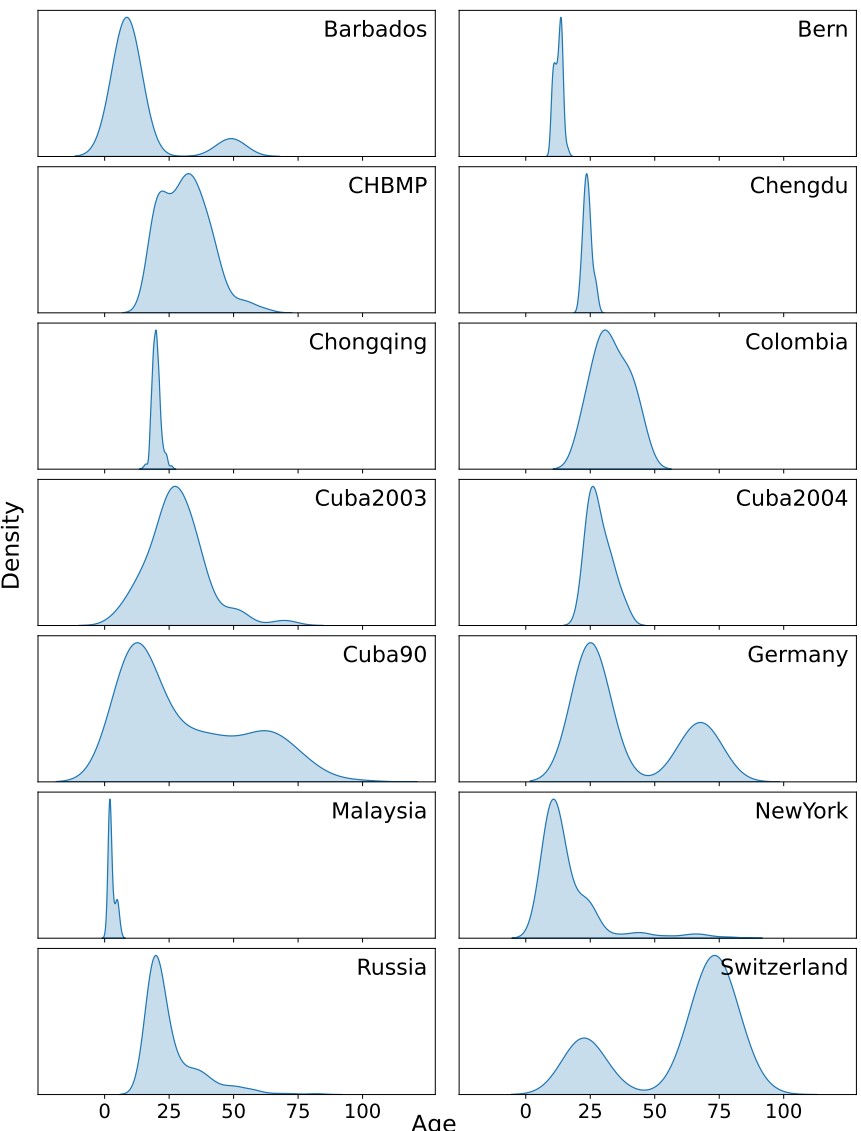

Figure 5: **Age distribution of the 14 sites of the HarMNqEEG dataset [33].** The distributions are represented with a kernel density estimate. The y-scales are not shared for visualization purpose.

## A.6 Table of performance scores.

Table 1: **Performance scores for different combinations of source sites.** The remaining sites were used as target domains.

Spearman's $\rho \uparrow$

| Sites source | DO Dummy | No DA | GREEN | Re-center | DO Intercept | GOPSA |
|---|---|---|---|---|---|---|
| Ba,Cho,G,S | $0.51 \pm 0.04$ | $0.63 \pm 0.02$ | $0.69 \pm 0.03$ | $0.52 \pm 0.02$ | $0.75 \pm 0.02$ | $\mathbf{0.78 \pm 0.02}$ |
| Be,Chb,S | $0.58 \pm 0.02$ | $0.73 \pm 0.01$ | $\mathbf{0.75 \pm 0.02}$ | $0.43 \pm 0.02$ | $0.68 \pm 0.02$ | $0.72 \pm 0.02$ |
| Ba,Co,G | $0.62 \pm 0.02$ | $0.64 \pm 0.02$ | $0.72 \pm 0.02$ | $0.42 \pm 0.02$ | $0.71 \pm 0.02$ | $\mathbf{0.74 \pm 0.02}$ |
| Cu03,M,R,S | $0.62 \pm 0.03$ | $0.63 \pm 0.01$ | $0.70 \pm 0.05$ | $0.46 \pm 0.02$ | $\mathbf{0.76 \pm 0.02}$ | $0.75 \pm 0.04$ |
| Ba,Be,Cho, Co,Cu90,G,R | $0.77 \pm 0.02$ | $0.79 \pm 0.01$ | $0.82 \pm 0.01$ | $0.44 \pm 0.03$ | $0.85 \pm 0.01$ | $\mathbf{0.87 \pm 0.01}$ |
| Mean | $0.62 \pm 0.03$ | $0.68 \pm 0.01$ | $0.74 \pm 0.03$ | $0.45 \pm 0.02$ | $0.75 \pm 0.02$ | $\mathbf{0.77 \pm 0.02}$ |

$R^2$ score $\uparrow$

| Sites source | DO Dummy | No DA | GREEN | Re-center | DO Intercept | GOPSA |
|---|---|---|---|---|---|---|
| Ba,Cho,G,S | $0.21 \pm 0.02$ | $0.06 \pm 0.06$ | $0.26 \pm 0.33$ | $-0.32 \pm 0.10$ | $0.57 \pm 0.03$ | $\mathbf{0.58 \pm 0.05}$ |
| Be,Chb,S | $0.25 \pm 0.02$ | $-0.07 \pm 0.08$ | $0.39 \pm 0.30$ | $-1.36 \pm 0.13$ | $0.43 \pm 0.03$ | $\mathbf{0.49 \pm 0.03}$ |
| Ba,Co,G | $0.48 \pm 0.03$ | $0.26 \pm 0.03$ | $0.47 \pm 0.09$ | $0.10 \pm 0.03$ | $0.60 \pm 0.03$ | $\mathbf{0.64 \pm 0.03}$ |
| Cu03,M,R,S | $0.26 \pm 0.02$ | $0.26 \pm 0.04$ | $0.48 \pm 0.13$ | $-0.30 \pm 0.07$ | $0.51 \pm 0.02$ | $\mathbf{0.51 \pm 0.09}$ |
| Ba,Be,Cho, Co,Cu90,G,R | $0.60 \pm 0.03$ | $0.54 \pm 0.02$ | $0.62 \pm 0.10$ | $0.14 \pm 0.02$ | $\mathbf{0.76 \pm 0.02}$ | $0.75 \pm 0.02$ |
| Mean | $0.36 \pm 0.03$ | $0.21 \pm 0.05$ | $0.44 \pm 0.19$ | $-0.35 \pm 0.07$ | $0.57 \pm 0.02$ | $\mathbf{0.59 \pm 0.04}$ |

MAE $\downarrow$

| Sites source | DO Dummy | No DA | GREEN | Re-center | DO Intercept | GOPSA |
|---|---|---|---|---|---|---|
| Ba,Cho,G,S | $9.25 \pm 0.16$ | $12.00 \pm 0.21$ | $9.08 \pm 1.98$ | $14.69 \pm 0.24$ | $7.69 \pm 0.19$ | $\mathbf{7.61 \pm 0.25}$ |
| Be,Chb,S | $9.48 \pm 0.14$ | $11.83 \pm 0.37$ | $8.67 \pm 2.30$ | $22.48 \pm 0.24$ | $9.28 \pm 0.20$ | $\mathbf{8.61 \pm 0.20}$ |
| Ba,Co,G | $9.42 \pm 0.14$ | $13.83 \pm 0.46$ | $9.44 \pm 0.77$ | $15.25 \pm 0.45$ | $8.77 \pm 0.15$ | $\mathbf{8.50 \pm 0.20}$ |
| Cu03,M,R,S | $9.64 \pm 0.17$ | $10.98 \pm 0.22$ | $\mathbf{8.53 \pm 1.12}$ | $16.50 \pm 0.25$ | $8.94 \pm 0.18$ | $8.75 \pm 0.72$ |
| Ba,Be,Cho, Co,Cu90,G,R | $10.37 \pm 0.23$ | $11.40 \pm 0.31$ | $9.53 \pm 1.10$ | $15.92 \pm 0.45$ | $8.53 \pm 0.23$ | $\mathbf{8.40 \pm 0.24}$ |
| Mean | $9.63 \pm 0.17$ | $12.01 \pm 0.31$ | $9.05 \pm 1.45$ | $16.97 \pm 0.33$ | $8.64 \pm 0.19$ | $\mathbf{8.37 \pm 0.32}$ |

## A.7 Figure 3 without normalization

Without `Re-center` and `Re-scale`:

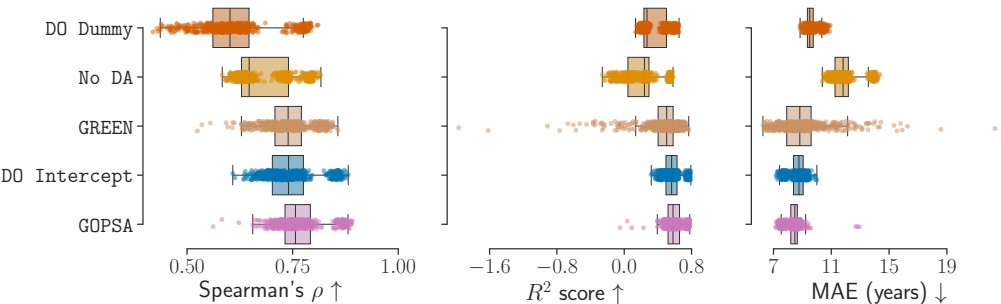

Figure 6: **Performance of four methods on several source-target combinations for three metrics:** Spearman's $\rho \uparrow$ (left), $R^2$ score $\uparrow$ (middle) and Mean Absolute Error $\downarrow$ (right). `Re-center` was removed from the plot to better visualize the other methods. A box represents the concatenated results across all site combinations. One point corresponds to one split of one site combination.

With `Re-center` and `Re-scale`:

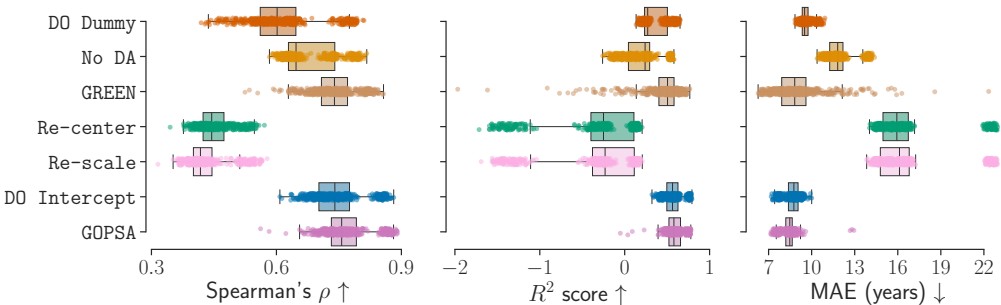

Figure 7: **Performance of all methods on several source-target combinations for three metrics:** Spearman's $\rho \uparrow$ (left), $R^2$ score $\uparrow$ (middle) and Mean Absolute Error $\downarrow$ (right). A box represents the concatenated results across all site combinations. One point corresponds to one split of one site combination.

## A.8 Boxplots of each source-target sites for the three metrics

The following figures represent the performance scores that are displayed in subsection A.6.

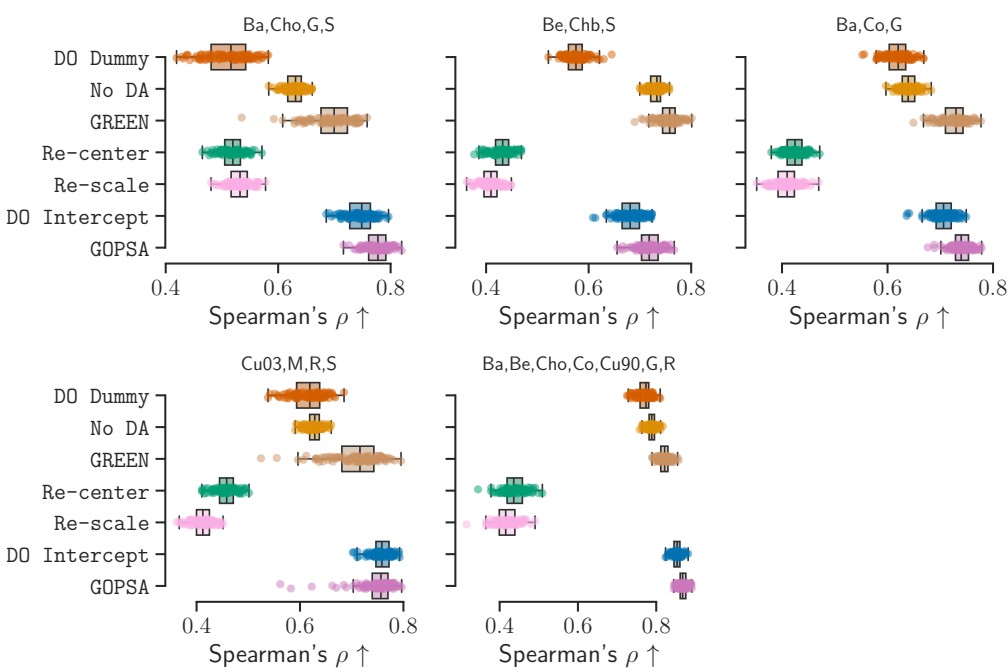

Figure 8: **Spearman's $\rho \uparrow$ for every site combination.** One panel corresponds to the results of one site combination. One point corresponds to one split.

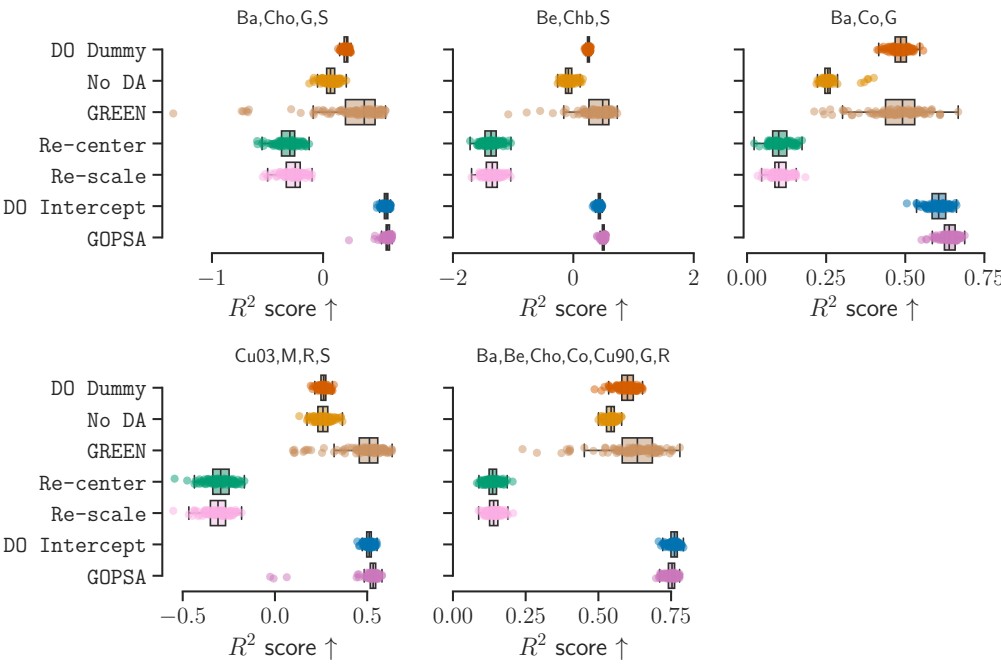

Figure 9: $R^2$ **score $\uparrow$ for every site combination.** One panel corresponds to the results of one site combination. One point corresponds to one split.

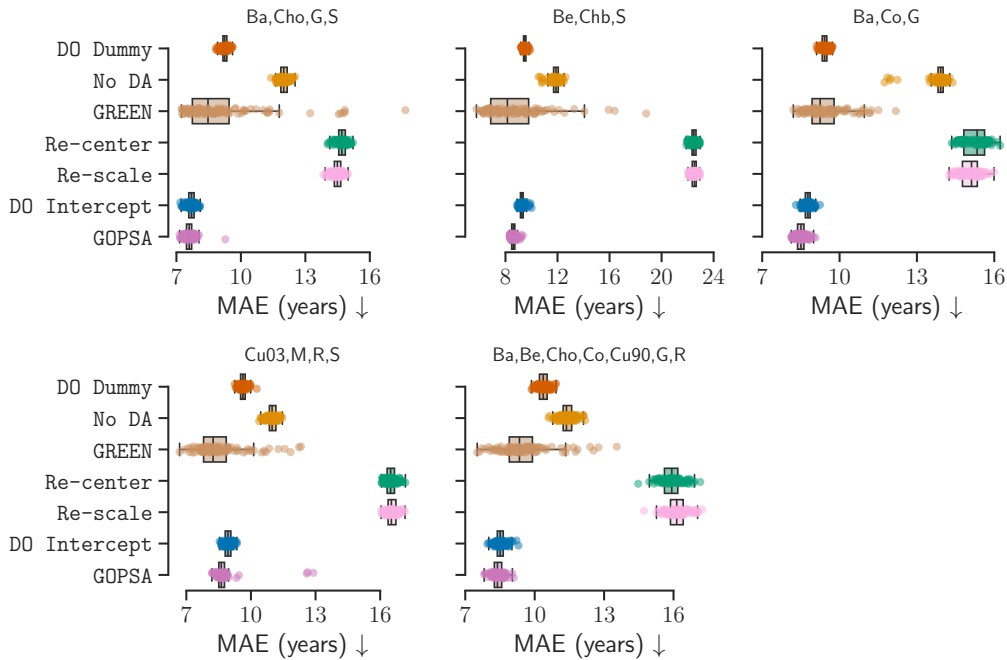

Figure 10: **Mean Absolute Error ↓ for every site combination.** One panel corresponds to the results of one site combination. One point correspond to one split.

