# OpenReview forum: "Geodesic Optimization for Predictive Shift Adaptation on EEG data"
_NeurIPS.cc/2024/Conference — NeurIPS 2024 spotlight_

### Official Review · Reviewer_r2hf · 2024-07-06

**Soundness:** 4
**Presentation:** 4
**Contribution:** 3
**Rating:** 7
**Confidence:** 4

**Summary:**

This paper proposes a novel method, Geodesic Optimization for Predictive Shift Adaptation (GOPSA), for predictive regression modeling with multi-source domain adaptation. The proposed method employs a domain-specific re-centering operator and a regression model using EEG data for age prediction. The method is innovative.

**Strengths:**

1. This paper proposes a novel riemannian-based solution with multi-source domain adaptation for a regression task.

2. Extensive experiments have been conducted, although only one dataset has been used.

3. I carefully checked the code and lemmas; the method, experimental design, stratified split (ensuring independence), and significance tests appear to be correct.

**Weaknesses:**

1. This method assumes knowledge of the average value of the target label $\bar y_T$, which is problematic due to the potential information leakage from the test set. As stated in lines 146-149, the method relies solely on $\bar y_T$ for adjusting regression model predictions, making it difficult to evaluate the generalizability of the method on unseen data.

2. The paper does not compare the proposed method with other alignment techniques such as re-scale and rotation correction [1].

3. There is no comparison with other Riemannian-based methods or state-of-the-art deep learning techniques for age prediction using M/EEG.

4. It may be inappropriate to claim 'achieved state-of-the-art performance' on a dataset without comparing it with other existing methods.



[1] Mellot, Apolline, et al. "Harmonizing and aligning M/EEG datasets with covariance-based techniques to enhance predictive regression modeling." Imaging Neuroscience 1 (2023): 1-23.

**Questions:**

The performance differences between the proposed GOPSA and the baseline, domain-aware (DO) intercept appear to be not large enough in several source-target site combinations. The T-test seems correct (100 repetitions per site combination). However, did you check the assumptions of the T-test: (1) whether the differences follow a normal distribution and (2) whether the variances are equal?

**Limitations:**

Only one dataset was used in this study. Experiments on different datasets should be conducted to validate the generalizability of the method.

---

> ### Author Rebuttal · Authors · 2024-08-07
>
> Dear Reviewer r2hf,
>
> We thank you for your detailed review. We took into account your comments and modified the experiments accordingly:
>
> **Addressing Weaknesses:**
> 1. **Knowledge of the target label mean:**\
> We acknowledge your concern regarding potential information leakage due to the assumption of knowing the target label mean value. We now estimate target $\bar{y}$ on target splits (50% of the data) that do not overlap with the evaluation target splits (50% of the data), rather than assuming target $\bar{y}$ to be known, ensuring no leakage from the test set. The revised Figure 2, included in the rebuttal PDF, shows that the performance remains comparable to when target $\bar{y}$ is assumed to be known.\
> We emphasize that target $\bar{y}$ is the only quantity computed from the target y. In many scenarios, it is acceptable to assume that target $\bar{y}$ is known, such as when a hospital has prior knowledge about its patient population.
> 2. **Comparison with the state-of-the-art:**\
> The Riemannian method “No DA” we use is already a strong baseline from recent works on regression from SPD matrices [e, f]. In response to your review, we have added “GREEN” [b], a recently proposed deep-learning method for biomarker prediction, to our experiments. We utilized the “g2” variant, which is an SPD network [d], since the data are SPD matrices. The revised Figure 2 shows that “GREEN” performs better than “No DA,” but still falls short of “GOPSA”. Although “GOPSA” shows superior performance, it is worth noting that the learned parallel transport could be integrated into “GREEN,” which we plan to explore in future work.\
> We also included the “Re-scale” baseline [a, f] in our experiments on real data. This method corrects second-order statistics on the SPD manifold. However, as shown in the revised Figure 2, this additional baseline suffers from similar issues as “Re-center” and does not effectively resolve the problem of joint shifts in (X, y). Additionally, [f] studied rotation corrections but found that a meaningful method could only be derived where data must be paired between domains (i.e., the same patients are present in several domains), which is not feasible in our experimental setup.\
> Overall, the revised Figure 2 clearly indicates that GOPSA remains the best-performing method compared to all tested methods.
>
> **Addressing Questions:**
> 1. **The performance difference between “GOPSA” and “DO Intercept”:**\
> From the revised Figure 2, we highlight that “GOPSA” significantly outperforms the “DO Intercept” which is the best performing baseline. Indeed, “GOPSA” significantly improved Spearman’s rank in 4 out of 5 site combinations (t-test p-values below 0.001). The results for the R² score and MAE are more nuanced, but on the majority of splits, both scores tend to be improved with “GOPSA” compared to “DO Intercept.”
> 2. **Statistical assumptions of the t-test:**\
> We thank you for your remark regarding the assumptions of the t-test. Firstly, we recall that we revised the computation of the target $\bar{y}$, leading to some changes in the p-values from the initial submission. To address your concerns, we examined the normality assumption of the score differences presented in the revised Figure 2B. Although space constraints prevented us from including Q-Q plots, our analysis confirms that the score differences generally follow normal distributions. Exceptions were noted for the MAE and R² scores in the site combination of the first row (Ba, Cho, G, S) and all scores in the site combination of the fourth row (Cu03, M, R, S). We also computed variances of scores of “GOPSA” and “DO Intercept” and obtained that they are equal except in the aforementioned cases. In these cases, the reported p-values are above 0.05, indicating that the score differences might not be statistically significant from zero and thus we do not mislead the reader. However, with more careful testing (without normal assumptions), some of the p-values could potentially be lower. We appreciate your attention to this detail, as it helped us ensure the robustness and reliability of our findings.
>
> **Addressing Limitation:**
> 1. **Dataset concerns:**\
> We use the multi-centric dataset HarMNqEEG (2022) [h], which effectively serves as a group of datasets by combining measurements from various hospitals, with data from over 1,500 subjects. This dataset encompasses varying conditions of measurements, including data from 9 countries, 14 studies, and 12 different EEG devices. This combination of heterogeneous datasets explains the presence of a (X, y) shift between sites. Consequently, testing across different combinations of sites allows us to simulate experiments across various measurement conditions effectively. This approach enables us to treat these site combinations as distinct datasets, thus providing a robust assessment of the generalizability of our method.\
> For future work, we welcome any recommendations for additional datasets that could further validate the generalizability and robustness of our method.
>
> [a] Rodrigues, P. L. C., et al. (2018). Riemannian Procrustes analysis: transfer learning for brain-computer interfaces. IEEE Transactions on Biomedical Engineering
>
> [b] Paillard, J., et al. (2024). GREEN: a lightweight architecture using learnable wavelets and Riemannian geometry for biomarker exploration. bioRxiv
>
> [d] Huang, Z., & Van Gool, L. (2017, February). A Riemannian network for spd matrix learning. In Proceedings of the AAAI conference on artificial intelligence (Vol. 31, No. 1).
>
> [e] Sabbagh, D., et al. (2019). Manifold-regression to predict from MEG/EEG brain signals without source modeling. Advances in Neural Information Processing Systems, 32.
>
> [f] Mellot, A., et al. (2023). Harmonizing and aligning M/EEG datasets with covariance-based techniques to enhance predictive regression modeling. Imaging Neuroscience, 1, 1-23.
>
> [h] Li, M., et al (2022). Harmonized-multinational qEEG norms (HarMNqEEG). NeuroImage, 119190.

---

> > ### Comment · Reviewer_r2hf · 2024-08-12
> >
> > I would like to thank the authors for all the additional experiments and the effort they put into addressing most of my concerns, as well as those of the other reviewers. Although there was not enough time to include an additional dataset to further test the generalizability, I am willing to increase the rating from 5 to 7.

---

### Official Review · Reviewer_ucmS · 2024-07-11

**Soundness:** 3
**Presentation:** 4
**Contribution:** 2
**Rating:** 7
**Confidence:** 3

**Summary:**

This paper presents a method for tackling domain adaptation challenges in EEG data analysis, specifically addressing shifts in both the feature space (represented by SPD matrices) and outcome variables $y$. The proposed method is designed on top of Riemannian mixed-effects model and is tailored for regression problems. Also, a key highlight of the proposed method is its capability to generalize from the source domain to any target domain without the need for retraining.

**Strengths:**

The paper is well-written at the beginning, with a clear introduction to concepts such as EEG data variability and its intuitive effect visualization, as in Figure 1. The paper provides clear theoretical foundations for establishing Riemannian geometry. The proposed method shows good performance improvement over baselines.

**Weaknesses:**

* Motivation Clarity.
My main concern is that the necessity of investigating domain adaptation (DA) methods on the SPD manifold is not convincingly presented. While SPD manifold representations are prevalent in EEG analysis, I did not see what specific issues were caused by domain shift. Or what unique issues does DA on the SPD manifold present that the proposed method wants to investigate and resolve? A clear problem setup is helpful for understanding the position of the research.

* The proposed method is on top of a Riemannian mixed-effects model (lines 142-143), which has been used to tackle data shifts in $X$ and $y$ as introduced in Related work (lines 68-74). While the authors mention that there is an opposing setting in previous studies (lines 75-78), it is unclear which specific limitation this paper focuses on and what the challenge is.

* The main characteristics/advantages of the proposed method are less explained. The paper overall explains very well about the derivation of the algorithm, but it is hard to understand the technical significance compared to existing methods and in what aspects it is better. While the work offers a thoughtful generalization of Riemannian mixed-effects model, it seems like it does not include any concepts that are wholly new.

* It would be helpful to elaborate on why domain shifts occur in both the data and biomedical variables. A minor question is: Is this an EEG-specific issue or common in biomedical data?

**Questions:**

Please refer to the weakness section.

**Limitations:**

I understand this is a professional paper on EEG study. However, I am currently unclear about its overall significance, particularly regarding the motivation or research question for investigating DA methods on the SPD manifold. However, my final decision is open to change pending the authors' rebuttal and further discussions.

---

> ### Author Rebuttal · Authors · 2024-08-07
>
> Dear Reviewer ucmS,
>
> We thank you for your detailed review of our paper. We considered your feedback and addressed the concerns and suggestions you raised to improve the clarity and impact of our research.
>
> **Clarification of Motivation and Problem Setup:**
> 1. **Motivation for domain adaptation on the SPD manifold:**\
> The necessity for domain adaptation (DA) methods is illustrated by the “No DA” approach, as detailed in the paper. This baseline corresponds to a SOTA covariance-based regression pipeline [e, f] without any adaptation between domains. As shown in Figure 2 from the PDF rebuttal, the “No DA” approach performs worse than a dummy model returning the average y per domain in terms of R² and MAE, demonstrating the critical need for adaptation techniques.\
> Classical domain adaptation methods on the SPD manifold are the “Re-center” and “Re-scale” methods [a, f] which correct shifts in X between domains. However, they exhibit very poor performance when both distributions in X and y vary between domains, as shown in Figure 2.\
> This joint shift issue on EEG data recently emerged with the ‘HarMNqEEG’ dataset [h], introduced in 2022, which we use in our experiments. Thus, this submission aims at presenting this problem and providing a new method called “GOPSA” to solve it.
> 2. **Riemannian mixed-effects models:**\
> In the existing literature (see e.g. [g]), Riemannian mixed-effects models typically assume that X belongs to a Euclidean space while y is in a Riemannian manifold. Our paper considers the opposite scenario, with X on a Riemannian manifold and y as a real-valued response. To the best of our knowledge, we are the first to address this problem, highlighting the novelty and significance of our approach. Thus, the proposed method is not made “on top of a Riemannian mixed-effects model” but the first to address mixed effects with X on a Riemannian manifold and y real-valued.
> 3. **Advantages of  GOPSA:**\
> We acknowledge the reviewer’s concern regarding the need for a clearer explanation of the main advantages of our method, “GOPSA”. To address this, we have added numerical experiments on simulated data that demonstrate how “GOPSA” effectively handles joint shifts in both X and y. The results are presented in Figure 1 of the rebuttal PDF. First, we leverage the classical instantaneous mixing model:
> $$x_i(t) = A\eta_i(t) $$
> where $x_i$ is the observed time-series, $i$ is the underlying signal of the neural generators and $A$ is the mixing matrix whose columns are the observed patterns of the neural generators. Furthermore, we use the log-linear model proposed in [e, f]
> $$y_i = \beta_0  + \sum_{\ell=1}^d \beta_{\ell} \log(p_{\ell i})$$
> where $p_{\ell i}$ is the variance of the $\ell$-th element of the underlying signal $\eta_i$. From this, we generate domains by applying two shifts. One on X that changes the mixing matrix per domain,
> $$x_i \mapsto B_k^{\xi} x_i$$
> with $B_k$ a domain-specific SPD matrix, $k$ the domain number and $\xi$ the variable that controls the intensity of the shift. A second shift is applied on y by shifting the variances per domain,
> $$p_{\ell i} \mapsto p_{\ell i}^{(1+k\xi)}.$$
> It should be noted that $\beta$ is kept the same across the domains.\
> The results are presented in Figure 1 of the rebuttal PDF. First, it shows that, if there is no shift in X, then "No DA" perfectly estimates the y because the log-linear model is respected across domains even when the y distribution changes (Figure 1B). However, the clear advantage of “GOPSA” is to estimate this log-linear model with shifts in (X, y) per domain (Figure 1C) which other methods can not do.\
> These experimental results on simulated results are consistent with the results on real data: “GOPSA” is the best method to estimate a shift on X while domains have different distributions in y.
> 4. **Domain shifts in biomedical data:**\
> Domain shifts in X often occur in biomedical data due to variations in data collection methods, such as differences in equipment, electrode placement, and environmental conditions. These factors can lead to inconsistent signal characteristics across different study sites. Shifts in y, the outcome variable, can result from demographic and clinical variations in the populations being studied, leading to differences in response distributions. While these issues are prominent in EEG data, they are common across various biomedical fields, necessitating robust adaptation techniques like our “GOPSA” method to ensure accurate cross-domain predictions.
>
> [a] Rodrigues, P. L. C., Jutten, C., & Congedo, M. (2018). Riemannian Procrustes analysis: transfer learning for brain-computer interfaces. IEEE Transactions on Biomedical Engineering, 66(8), 2390-2401.
>
> [e] Sabbagh, D., Ablin, P., Varoquaux, G., Gramfort, A., & Engemann, D. A. (2019). Manifold-regression to predict from MEG/EEG brain signals without source modeling. Advances in Neural Information Processing Systems, 32.
>
> [f] Mellot, A., Collas, A., Rodrigues, P. L., Engemann, D., & Gramfort, A. (2023). Harmonizing and aligning M/EEG datasets with covariance-based techniques to enhance predictive regression modeling. Imaging Neuroscience, 1, 1-23.
>
> [g] Schiratti, J. B., Allassonniere, S., Colliot, O., & Durrleman, S. (2015). Learning spatiotemporal trajectories from manifold-valued longitudinal data. Advances in neural information processing systems, 28.
>
> [h] Li, M., et al (2022). Harmonized-multinational qEEG norms (HarMNqEEG). NeuroImage, 256, 119190.

---

> > ### Comment · Reviewer_ucmS · 2024-08-12
> >
> > Thank you to the authors for the response and efforts. The additional experiments improve the quality of the work and address my concerns. I am increasing my rating to 7 and hope that the motivation behind the study will be more clearly presented in the final version.

---

### Official Review · Reviewer_N9Jm · 2024-07-13

**Soundness:** 3
**Presentation:** 3
**Contribution:** 3
**Rating:** 8
**Confidence:** 4

**Summary:**

The authors proposed GOPSA a new approach for alignment of EEG datasets from multiple subjects and sites. The method respects  the Riemannian manifold of the covariance matrices and learns the parallel transport length parameter simultaneously with the regression model used for solving the downstream task. While simple  and intuitive the approach is well justified, rooted in the previous research and  adequately evaluated using a large public EEG dataset and the associated age prediction task.

**Strengths:**

1. The method is interesting, simple, uses few parameters and as the authors demonstrated efficient.
2. The method is well described with all necessary details to comprehend it and implement.
3. The dataset is large and comprehensive and extensive comparison with other relevant data alignment techniques is presented.

**Weaknesses:**

1. No performance analysis on a simulated data
2. No interpretation of the obtained decision rule.
3. Table 1 - not all the difference in performance of GOPSA with the other competing techniques seem significant.

**Questions:**

1. Could the authors provide some interpretation of the age prediction model? What frequency bands appear pivotal? Which electrodes?
2. Could the authors add the statistical test to Table 1?
3. Could the authors discuss and compare their approach against the technique used in several DL studies applied to the multisubject datasets when a specific subject adaptation trainable layer is used to interface  his or her data with the "oracle" classification engine?
4. The authors use covariance matrices as features, however it seems that the transport learnt can be applied to the actual multichannel data directly - could the authors discuss this possibility in the light of forming large aligned datasets in the channel x time form rather than their covariance matrices?

**Limitations:**

The authors adequately addressed the limitations of their study, see also Weaknesses section of this review.

---

> ### Author Rebuttal · Authors · 2024-08-07
>
> Dear Reviewer N9Jm,
>
> Thank you for your positive assessment of our submission. In the following, we detailed answers to the weaknesses and questions you raised.
>
> **Addressing Weaknesses:**
> 1. **Evaluation of the methods on simulated scenarios:**\
> We agree with the reviewer and added numerical experiments on simulated data to illustrate the advantage of “GOPSA” compared to other methods. All methods are evaluated in 3 simulation scenarios: shift in X only, shift in y only, and joint shift in X and y. The results are presented in Figure 1 of the attached PDF file. We demonstrate that “GOPSA” effectively compensates for the joint shift in both X and y, and is robust to shifts in only X or only y.
> 2. **Interpretability of Riemannian models:**\
> The interpretation of Riemannian-based models, such as the one we refer to as “No DA,” involves transforming the model’s parameters into interpretable patterns, as demonstrated in prior work [c]. Applying similar techniques to “GOPSA” could yield valuable insights into its decision-making process, revealing the patterns and features that contribute most significantly to its predictions. This analysis is left for future research, and could enhance the transparency and interpretability of our method and guide further refinements and applications.
> 3. **Statistical significance of the results:**
> The reviewer is right that the difference in performance between “GOPSA” and other techniques is not always significant. The previously mentioned analysis done on simulated data showed that “GOPSA” clearly outperforms other techniques in the presence of a joint shift in X and y. Thus, with real data, we expect “GOPSA” to perform better when there is a joint shift in X and y, and to perform similarly to other methods when it is not the case. In Figure 2, which we updated and included in the attached file, we report the p-values of a t-test on the difference “GOPSA” minus ”DO Intercept” for three metrics on all site combinations. We chose to perform this test with “DO Intercept” as it is the competing method with the best performances. “GOPSA” significantly improved Spearman’s rank in 4 out of 5 site combinations. The results for R2 score and MAE are more nuanced, but on the majority of splits both scores tend to be improved with “GOPSA”.
>
> **Addressing Questions:**
> 1. **Interpretation of the age prediction model:**\
> As explained previously, we agree with the reviewer that an interpretation analysis of the model would be interesting. In this work we mainly focused on the development of a novel approach to the specific setting of joint shift in both X and y. In our opinion, the specific work required to perform a clean and thorough interpretation represents a contribution in itself, see e.g. [c]. Indeed, the interpretation is not straightforward and would require taking into account many factors: age and frequency are linked, and confounding factors, like the sexe and the mental health of the participants.
> 2. **Statistical test results:**\
> A statistical t-test was conducted to compare the performance of “GOPSA” with that of the best-performing baseline method, “DO Intercept”. The p-values resulting from this test are presented in Figure 2 and have been updated in the rebuttal PDF. The statistical analysis from Figure 2 ensures a rigorous evaluation of the significance of the performance differences between “GOPSA” and the baselines. “GOPSA” significantly improved Spearman’s rank in 4 out of 5 site combinations. The results for R2 score and MAE are more nuanced, but on the majority of splits both scores tend to be improved with “GOPSA”. Regarding Table 1, we agree that including p-values would be beneficial. However, due to space constraints, we will add them in a subsequent version of the manuscript.
> 3. **Comparison with DL model:**\
> We added in the benchmark a deep learning (DL) approach, called “GREEN”, that has recently been developed for EEG applications like age prediction [b]. We utilized the “g2” variant, which is an SPD network [d], because the data are SPD matrices. Even though the architecture does not include any adaptation layer, the results on simulated (Figure 1 of the rebuttal PDF) and real data (Figure 2) showed this model to be relatively robust to shifts. On real data “GREEN” performed better than “No DA”, the classical Riemannian-based regression pipeline but  it still falls short of the proposed “GOPSA” method which is tailored for the studied problem. However, as mentioned in the conclusion of the paper, since “GOPSA” is implemented using PyTorch, a future perspective would be to integrate “GOPSA” into Riemannian DL models like “GREEN”.
> 4. **Application of the transport to the EEG time series:**\
> In this work, the dataset we used did not provide the raw EEG time series, so we focused on covariance matrices as features. However, the reviewer’s remark is correct, and the transportation operators learned from the covariance representation could be applied to the EEG time series. We think that it would be interesting to investigate this approach with datasets that provide the raw EEG signals.
>
> [b] Paillard, J., Hipp, J. F., & Engemann, D. A. (2024). GREEN: a lightweight architecture using learnable wavelets and Riemannian geometry for biomarker exploration. bioRxiv, 2024-05.
>
> [c] Kobler, R. J., Hirayama, J. I., Hehenberger, L., Lopes-Dias, C., Müller-Putz, G. R., & Kawanabe, M. (2021, November). On the interpretation of linear Riemannian tangent space model parameters in M/EEG. In 2021 43rd Annual International Conference of the IEEE Engineering in Medicine & Biology Society (EMBC) (pp. 5909-5913). IEEE.
>
> [d] Huang, Z., & Van Gool, L. (2017, February). A Riemannian network for SPD matrix learning. In Proceedings of the AAAI conference on artificial intelligence (Vol. 31, No. 1).

---

### Official Review · Reviewer_85Rg · 2024-07-13

**Soundness:** 4
**Presentation:** 4
**Contribution:** 4
**Rating:** 8
**Confidence:** 5

**Summary:**

This study presents an approach to learning robust models under joint shifts in X and y (an important issue in healthcare). Focusing on EEG signals, a covariance-matrix-based learning framework is developed to address this challenge. Empirical results on EEG-specific benchmarks demonstrate its overall efficacy.

**Strengths:**

- Focus on a highly relevant and unaddressed problem in EEG data analysis.
- The proposed solution is innovative (mixed effects modeling w/ representation learning) and scalable (it does not need model retraining or access to source data).
- Clear technical exposition/prose.

**Weaknesses:**

- Lack of discussion/comments on: 1) other relevant solutions for the same problem setting since joint shifts in X and y are an issue for many other biosignal modalities, 2) expected failure modes and future directions based on this approach.

**Questions:**

- Q: Is the ComBat harmonization tool (used in imaging and genetics to correct batch effects) a relevant solution within the same problem setting? "ComBat is a batch adjustment method that removes additive and multiplicative differences between sites due to the use of different scanning devices" - https://cran.r-project.org/web/packages/combat.enigma/index.html.
- Q: What happens when source and target distributions for the response variable don't overlap or only partially? (regarding the choice made in line 238).
- Q: What happens if we only have a bad/noisy estimate of the target domain's mean value (in practice)? Is there any intuition on how things would break?

**Limitations:**

See weaknesses.

---

> ### Author Rebuttal · Authors · 2024-08-07
>
> Dear Reviewer 85Rg,
>
> Thank you for your positive evaluation of our submission. We took into account your concerns, and answered your questions point by point in the following:
>
> **Addressing Weaknesses and Questions:**
> 1. **Other similar problem settings and Combat harmonization algorithm:** \
> The studied problem setting is indeed present in other biomedical applications. For example, you mentioned ComBat which was first developed for gene expression analysis. Other linear mixed-effect models are also usually applied to biomedical data in this context. However, they are often used to harmonize data to perform statistical analysis and thus are not adapted for predictive applications of machine learning. Indeed, to the best of our knowledge, mixed-effects models, like ComBat, assume to have access to the biomedical outcome (y) of source and target domains to compute corrections. In the studied setting, we assume to only have access to the y mean of each target site which is a much harder problem. Furthermore, the proposed benchmark includes the DO Intercept method, which is well suited for machine learning applications, as an alternative to linear mixed-effect models.
> 2. **Overlap of y distributions between source and target domains:**\
> The second question raises an important point. Theoretically, based on the generative model of the simulated data (as described in the caption of Figure 1 in the rebuttal PDF), the data and outcome (y) are linked by a log-linear relationship. This implies that, knowing the shift in X for the target domain, predictions can be made even when y distributions do not overlap between the source and target. Since GOPSA estimates the target shift in X by minimizing $(\bar{y} - \hat{y}_i)^2$, it is capable of handling such scenarios.
> 3. **Noisy estimate of the target $\bar{y}$ :**\
> First, we want to point out to the reviewer that we changed how the target $\bar{y}$ is estimated in our empirical benchmark on real data.  Indeed, we estimate target $\bar{y}$ on splits (50% of the data) that do not overlap with the evaluation target splits (50% of the data), rather than assuming target $\bar{y}$ to be known. This leads to noisier target $\bar{y}$ estimates. We noticed that the results are similar to the submission’s one. We still note that the performances of 3 splits (out of 100) in site combination 4 (Cu03, M, R, S) have deteriorated. Figure 2 of the rebuttal PDF presents the updated results.\
> Our intuition is that the noisier the target $\bar{y}$ estimation is, the worse the performance will get to the point of suppressing the interest in applying a domain-specific adaptation. A future perspective of this work is to study the point at which the adaptation does more harm than good.

---

> > ### Comment · Reviewer_85Rg · 2024-08-12
> >
> > Thank you authors for the rebuttal and taking the effort to run additional experiments. I think the points in this rebuttal and others would add value for readers in the discussion and future work sections of the final version. I maintain my initial recommendation.

---

### Author Rebuttal · Authors · 2024-08-07

We thank the reviewers for the thorough and insightful reviews of our submission.

We appreciate the acknowledgment of our method’s novelty, simplicity, and effectiveness in addressing domain adaptation challenges in EEG analysis. We are also glad that the presentation and technical soundness were noted positively across the reviews.

To address the weaknesses of the paper, we submitted a PDF file in this rebuttal with two figures:
- Figure 1 is new and contains simulated experiments where shifts are applied on either X, y, or both (X, y). These experiments clearly demonstrate the efficiency of the proposed method, “GOPSA”, in estimating shifts in X between domains, even in the presence of a shift in y, contrary to the baselines. This additional evaluation of domain-adaptation methods via ground-truth simulation of the data-shift generating process should help the reader to build intuition about the studied problem and the added value of the proposed solution “GOPSA”. A new section with these simulated data will be added to a subsequent version of the paper.
- Figure 2 is a revision of the Figure 2 of the submission. First, we estimate target $\bar{y}$ on target splits that do not overlap with the evaluation target splits, rather than assuming target $\bar{y}$ to be known. Second, we included two additional baselines in the experiments on real data: “Re-scale” [a], which corrects second-order statistics on the SPD manifold, and “GREEN” [b], a deep-learning architecture tailored for EEG data. “Re-scale” performs similarly to “Re-center”, i.e., is worse than all other methods. “GREEN” performs better than “No DA” but is still far from “GOPSA,” which is tailored for the studied problem. Overall, “GOPSA” still outperforms the strongest baseline, “DO Intercept.” Indeed, “GOPSA” significantly improved Spearman’s rank in 4 out of 5 site combinations (t-test p-values below 0.001). The results for the R² score and MAE are more nuanced, but on the majority of splits, both scores tend to be improved with “GOPSA” compared to “DO Intercept.” These findings underscore GOPSA’s superior performance and robustness in addressing the challenges of joint shifts in X and y.

In addition, we would like to emphasize that “GOPSA” was specifically developed to handle joint shifts in the data distribution and the outcome distribution, as illustrated by the simulations in Figure 1 of the rebuttal file. We thus expect “GOPSA” to outperform the baseline methods (e.g. “DO Intercept”) whenever joint (X, y) shifts occur. In our experimental benchmark, “GOPSA” significantly outperformed the baseline methods in some site combinations, but not all. This allows us to assume that not all site combinations show joint shifts. We, therefore, argue that this signature of “GOPSA” outperforming “DO Intercept” can be used by researchers as a decision rule to infer the presence of joint shifts and, hence, can serve as a tool for data exploration and model interpretation.

Finally, point-by-point answers to each reviewer are provided with respect to the different reviews.

[a] Rodrigues, P. L. C., Jutten, C., & Congedo, M. (2018). Riemannian Procrustes analysis: transfer learning for brain-computer interfaces. IEEE Transactions on Biomedical Engineering, 66(8), 2390-2401.

[b] Paillard, J., Hipp, J. F., & Engemann, D. A. (2024). GREEN: a lightweight architecture using learnable wavelets and Riemannian geometry for biomarker exploration. bioRxiv, 2024-05.

---

### Decision · Program_Chairs · 2024-09-25

**Decision:**

Accept (spotlight)

**Comment:**

This paper presents a method for tackling domain adaptation challenges in EEG data analysis, specifically addressing shifts in both the feature space (represented by SPD matrices) and outcome variables. The approach is novel and well justified. The experimental evaluation seems not sufficient in the initial reviews, and the authors' rebuttal with additional experiments has successfully addressed this issue. The motivation of the study should be more clearly presented in the final version.